# Arithmetic optimization based MPPT for photovoltaic systems operating under nonuniform situations

Maheshwari Adaikkappan[1], Nageswari Sathiyamoorthy[2], Durga Devi Ravichandran[3], Karthikeyan Balasubramani[4], Sundararaju Karuppannan[5], Ramasamy Palanisamy[6], Zakaria M. S. Elbarbary[7], Saad F. Al-Gahtani[7], Ahmed I. Omar[8] *

1 Department of Electronics and Communication Engineering, M. Kumarasamy College of Engineering, Karur, Tamilnadu, India, 2 Department of Electrical and Electronics Engineering, Alagappa Chettiar Government College of Engineering and Technology, Karaikudi, Taminadu, India, 3 Department of Electrical and Electronics Engineering, B.S. Abdur Rahman Crescent Institute of Science and Technology, Chennai, Tamilnadu, India, 4 Department of Robotics and Automation Engineering, PSG College of Technology, Coimbatore, Tamilnadu, India, 5 Department of Electrical and Electronics Engineering, M Kumarasamy College of Engineering, Karur, Tamilnadu, India, 6 Department of Electrical and Electronics Engineering, SRM Institute of Science and Technology, Kattankulathur, Chengalpattu, Taminadu, India, 7 Electrical Engineering Department, College of Engineering, King Khalid University, Abha, Saudi Arabia, 8 Department of Electrical Power and Machines Engineering, Higher Institute of Engineering, El-Shorouk Academy, El-Shorouk City, Cairo, Egypt

* a.omar@sha.edu.eg

**Data Availability Statement:** All relevant data are within the manuscript.

## Abstract

Photovoltaic (PV) modules may encounter nonuniform situations that reduce their useable power volume, causing ineffective maximum power point tracking (MPPT). Moreover, due to the incorporation of bypass diodes, power-voltage (P-V) graph has multi-peaks when each component of the module receives different solar irradiation. This paper proposes a solution to this problem using an arithmetic optimization algorithm (AOA) for MPPT in PV systems operating in nonuniform situations. The non-operational regions associated with the voltage are excluded using a single-ended primary inductance converter (SEPIC) with voltage step-up and step-down capability. The AOA-MPPT algorithm gets current and voltage as inputs from the PV modules. It computes the converter's duty cycle and regulates the operational point to keep MPP under all working conditions. The proposed AOA-MPPT's efficacy under different insolation patterns has been validated using three nonuniform conditions in terms of convergence, tracking speed, steady state oscillations, and tracking efficiency. In simulations, the proposed AOA-MPPT method and SEPIC converter demonstrated quick response and excellent steady-state performance. The tracking efficiency of the AOA-MPPT is above 99% and settling time is 200 to 300ms for all three nonuniform conditions.

**Funding:** the authors extend their appreciation to the Deanship of Scientific Research at King Khalid University under for funding this work through General Research Project under Grant number (RGP2/425/44).

**Competing interests:** The authors have declared that no competing interests exist.

# 1. Introduction

With the huge capital cost of PV systems, it is essential to guarantee optimal output of accessible PV energy. It is accomplished either through enhanced cell design or through the use of MPPT's efficient power tracking technology [1, 2]. However, achieving optimal operation is frequently challenging owing to the non-linear characteristic curve of PV sources, which is degraded further in partial shade conditions. Power-voltage curves deform into a multi-peak shape but only one global peak (GP) in this state. As a result, PV systems must function at true GP; otherwise, considerable efficiency losses can be observed. MPPT can be roughly classified into three categories such as traditional MPPT methods, Machine learning based MPPTs and Optimized MPPTs [3, 4].

In the first category, typically two steps are used to precisely track the real Global Maximum Power Point (GMPP) of a PV array. To determine the localization of GMPP, the initial phase presents a sequence of significant perturbations in the control signal. The information acquired by these perturbations is adequate for the standard Perturb & Observe (PO) or Incremental Conductance (IC) approach, which is used in the second phase, to guarantee the operation at GMPP. In spite of producing relatively excellent outcomes, their key downsides are discovered to be limited speed owing to scanning of a whole I-V curve and added complication for customisation [1]. Furthermore, the conventional MPPTs may become trapped in a Local Maximum Power Point (LMPP), owing to the tracking features of these algorithms' inability to distinguish between the LMPP and the GMPP [5].

In machine learning based MPPT schemes, the concepts of Neural Network (NN) [6] and Fuzzy Logic (FL) are used to trace the GMPP. These MPPT approaches are undoubtedly effective; nonetheless, a few flaws are discovered. For NN, tracking precision is extremely reliant on the existing training information of the variables, which may not cover the entire day or entire month. Several processing phases are required in FL controller to calculate the final control output [1]. Accordingly, there must be a trade-off between tracking speed and calculation cost.

Considering GMPP tracking as an optimization problem, many researchers have recently used a number of optimization algorithms to find the optimal operational point of PV. Because the shaded properties of a PV are multi-modal in nature, an optimization algorithm appears to be a highly appropriate choice for MPPT application [1, 7]. This is obvious from various recent efforts on GMPP tracking, such as genetic algorithms, ant colony optimization [8], chaotic search [9], Particle Swarm Optimization [5, 10], grey wolf optimization (GWO) [11], cuckoo search, and differential evolution (DE) [1, 12], golden section search [13], invasive weed optimization [14], roach infestation optimization [15]. The results of the review made by Jordehi et al., [16] show that the best options for MPPT are metaheuristic optimisation algorithms because of their advantages, which include system independence, efficient performance in partial shading situations, and lack of oscillations around the maximum power point.

PV must work at MPP to get the extreme power under any irradiance and temperature conditions. Thus, it is conceivable to integrate a DC/DC converter with a computational system that will alter the duty cycle [17] and implicitly the input impedance of the converter according to the search strategy until the system reaches the MPP, overcoming undesirable impacts on output power [5]. Because of recent advancements in the efficiency of power electronics converters and effective algorithms, module level MPPT applications have made significant progress [18, 19]. This kind of MPPT technique enables PV units to work at high efficacy even in nonuniform working situations like partial shading.

As a result, the MPP tracker is coupled with a DC-DC converter to continuously ensure maximum power transfer [20]. It is critical to understand which DC-DC converter is appropriate for a specific state and how it works in order to have the optimum power transmission

[21]. A maximum power point tracker, in combination with a high gain converter, is used to track the fluctuating power [22, 23]. The converter is used to balance the power between a PV system and grid or a stand-alone system [24]. Therefore, choosing a converter is extremely significant for maximizing the system output and ensuring the overall operation's safety [25]. High voltage gain converters are very famous in the recent research as a means of integrating renewable energy sources [26, 27].

The converter serves as an interface, trying to balance power between a PV system and the grid or a stand-alone system. As a result, selecting a converter is critical for optimising system output and assuring the overall safety of the operation [28]. The two primary types of converters available are isolated and non-isolated [29]. Separation among the input and output sides of an isolated converter is accomplished by removing the DC path with a transformer. It is shielded from the high voltage passing through the DC-DC converter [30]. Most isolated converters have discontinuous input currents, rendering them unsuitable for solar energy applications. As a result, larger input filter capacitors would be required, increasing the size of the converter.

A non-isolated DC-DC converter has fewer components, is smaller in size, and has less energy loss because there is no separation or transformer [25]. In PV applications, the frequently used non-isolated converter, i.e., boost converter, achieves high gain by lengthening duty cycle [31–34]. High switching loss, saturation, reverse recovery problems, and low performance are the main downsides of boost converters [35]. Power converters must have a high gain while keeping a low duty cycle in order to be used in PV applications [36–38]. The buck-boost converter combines the advantages of the fundamental buck and boost converter topologies and has been utilized effectively in a PV application [39]. The buck-boost converter, on the other hand, is still being researched to improve the effectiveness of solar PV [32]. Researchers are evolving non-isolated buck-boost converters, such as Cuk for continuous output [40], single-ended primary inductance converter (SEPIC) [41, 42], and Luo converters for high gain [43].

Fig 1 depicts the standalone PV with MPPT driven SEPIC, which is employed in this article to track the MPP of a PV system. The PI controller is no longer present, and the MPPT algorithm directly computes duty cycle (control signal). This is commonly referred to as a direct control MPPT scheme in the literature. Simplified control loop, Calculation time reduction and exclusion for PI controller tuning are the advantages of direct control MPPT schemes. Eventhough the feedback loop is absent; the direct control technique achieves comparable optimal results. This article proposes a solar PV system which includes AOA-MPPT algorithm that tracks the MPP effectively to reduce the effect of dissimilarities due to the nonuniform climatic conditions. But in the majority of metaheuristic algorithms, even slight changes in load or solar irradiation can cause the system to exhibit excessive PV power fluctuations due to the methods used to achieve the GMPP [44, 45]. Thus, this work analyses the effects of modest load change, two different fast irradiance shifting circumstances, and three non-uniform insolation scenarios with different GP placements. Furthermore, maximum power achieved, convergence time [44] and tracking efficiency are noted.

The key objective of the article is,

➢ To design an efficient Arithmetic optimization algorithm based MPPT method and apply to a PV system with a SEPIC converter.

- It can track the extreme power under rapidly changed climate conditions and provide the continuous output with SEPIC converter.

- It limits the computational complexity and have a high convergence speed to increase the reliability.

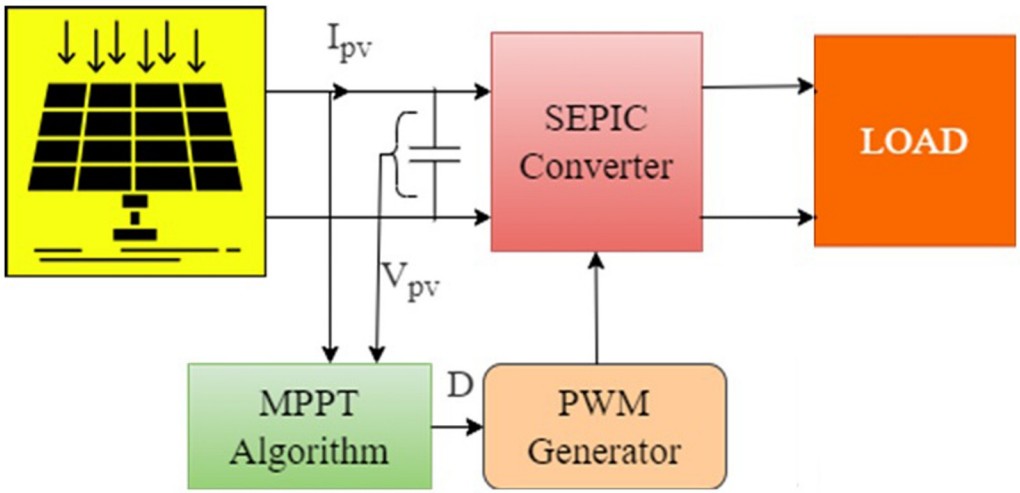

**Fig 1. Standalone PV with SEPIC.**

- The proposed system can effectively track the MPP and has no steady-state oscillation, as well as good dynamic performance.

The remaining of the article is set out as follows. Section 2 gives the features of PV array, and Section 3 gives the characteristics of SEPIC. Section 4 introduces the AOA algorithm for maximum power point tracking. Section 5 defines the simulation results of the AOA algorithm based MPPT with SEPIC converter and estimates the proposed method over nonuniform atmospheric conditions. The conclusion of the article is stated in Section 6.

## 2. PV characteristics

Fig 2 depicts a single diode equivalent circuit used to describe a PV cell [46]. Ideally, a single PV cell serves as a DC source in conjunction with an antiparallel diode. The irradiance intensity (G) is related to the output current ($I_{PV}$).

The model incorporates the effects of irradiance and temperature. In the ideal single diode model $I_{PV} = I_L$-$I_D$, where $I_L$ is the current generated by the incidence of light and $I_D$ is the current through the diode, as given in Eq (1). Intrinsic resistances are included in a practical model. Eq (2) provides a revised mathematical model. The intrinsic resistances and capacitances of junctions have a considerable impact on the model's behaviour.

$$I_{PV} = I_L - I_s\left[exp\left(\frac{V_{PV}}{\alpha V_T}\right) - 1\right] \tag{1}$$

$$I_{PV} = I_L - I_s\left[exp\left(\frac{V_{PV} + I_{PV}R_s}{\alpha V_T}\right) - 1\right] - \left(\frac{V_{PV} + I_{PV}R_S}{R_{Sh}}\right) \tag{2}$$

$$V_T = \frac{N_s k T}{q} \tag{3}$$

$$I_{PV} = I_L\,N_p - N_p I_s\left[exp\left(\frac{V_{PV} + I_{PV}R_s}{N_s \alpha V_T}\right) - 1\right] - \left(\frac{V_{PV} + I_{PV}R_s}{R_{Sh}}\right) \tag{4}$$

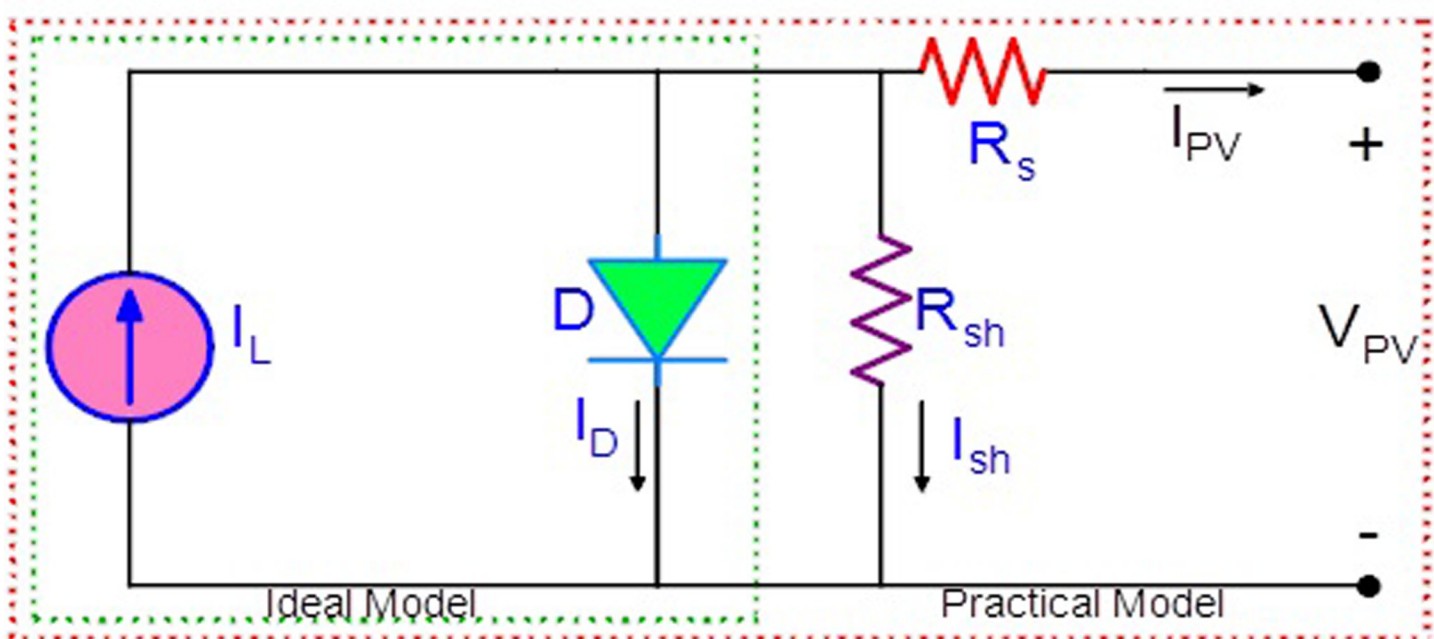

**Fig 2. Single diode PV model.**

This model represents the PV system in a conventional way. Eq (3) is adjusted to Eq (4) for $N_s$ cells arranged in series and $N_p$ cells linked in parallel (4). The impact of variation in irradiance and temperature on P-V and I-V curves are represented in Fig 3.

The total power of a PV array is the summation of the power from all of the modules that are arranged in series and/or parallel to make up the array. Fig 3 depicts a PV array covers four PV modules that are arranged in series. When one of the PV modules is sheltered, it functions as a load rather than a power supply. The shaded PV module will be damaged in the long run due to the hotspot's phenomena. As a result, bypass diodes are included to safeguard PV

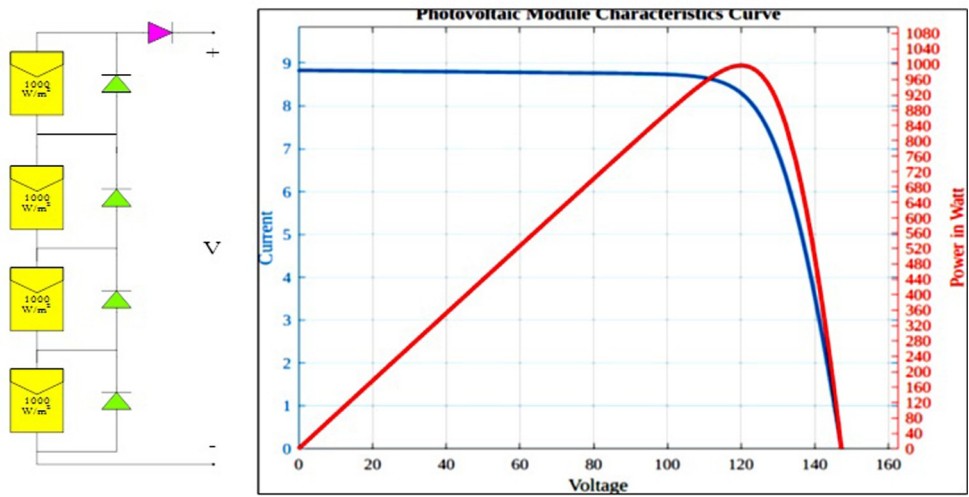

**Fig 3. PV array operation with uniform irradiation.**

modules from self-heating under partial shadowing. The bypass diodes are reverse biased and have no effect under uniform insolation.

When the PV module is in the shade, the diode across it is forward biased, and current flows over the diode rather than the PV module. Fig 4 illustrates how the diode alters the P-V arcs into a more distorted figure with several peaks. Therefore, the system should be adjusted to the GMPP in order to extract the most power out of the PV array. Up to 70% of the power could be lost if this is not the case [47]. Thus, an intelligent and effective MPPT approach should be applied to accomplish optimal energy harvesting from the PV array.

## 3. Converter characteristics

DC-DC converters are commonly utilised in standalone PV structures for MPPT [48–51]. As a result, it is critical to demonstrate the MPPT capabilities of converters in PV systems. The SEPIC converter has buck and boost modes. As a result, it is able to monitor the MPP at various power level. SEPIC has numerous advantages over other buck-boost converter topologies in terms of execution and MPPT ability. The circuit diagram for the SEPIC is depicted in Fig 5. Despite having two inductors and capacitors that lengthen the dynamic reaction time, unlike buck-boost converters, these converters do not need extra circuits for switch driving.

The following (5)-(8) is a list of the differential equations for SEPIC.

$$L_1 \frac{dI_{L1}}{dt} = -(1-k)(V_{C1} + V_{C2}) + V_{PV} \tag{5}$$

$$L_2 \frac{dI_{L2}}{dt} = kV_{C1} - (1-k)V_{C2} \tag{6}$$

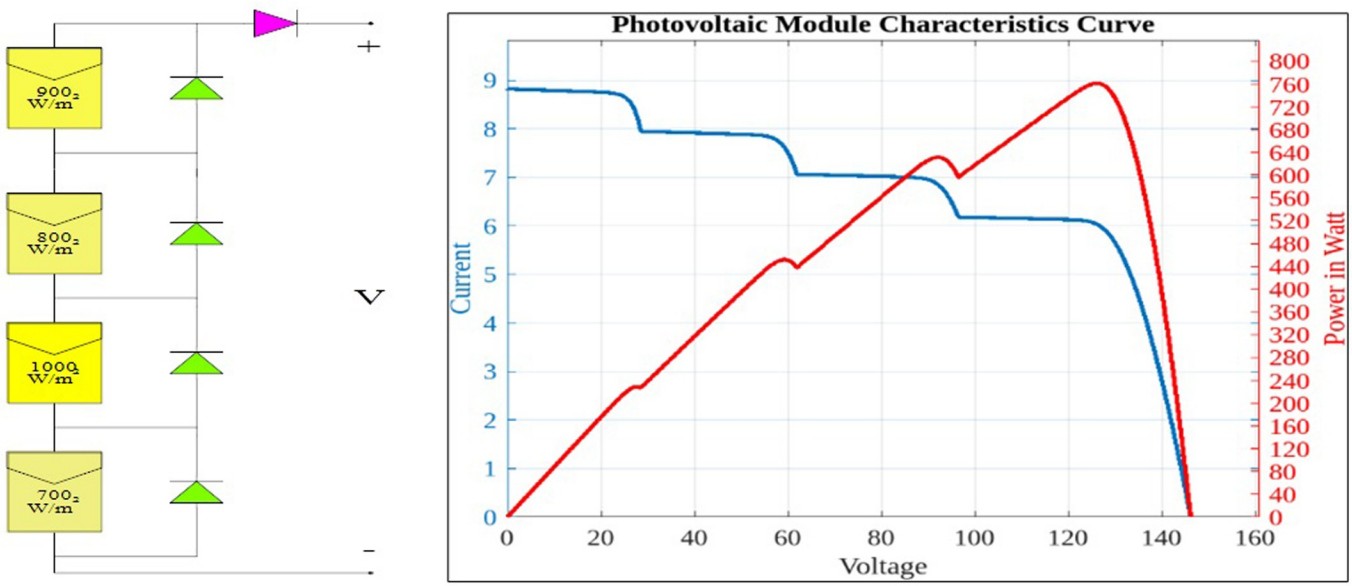

**Fig 4. PV array operation with nonuniform condition.**

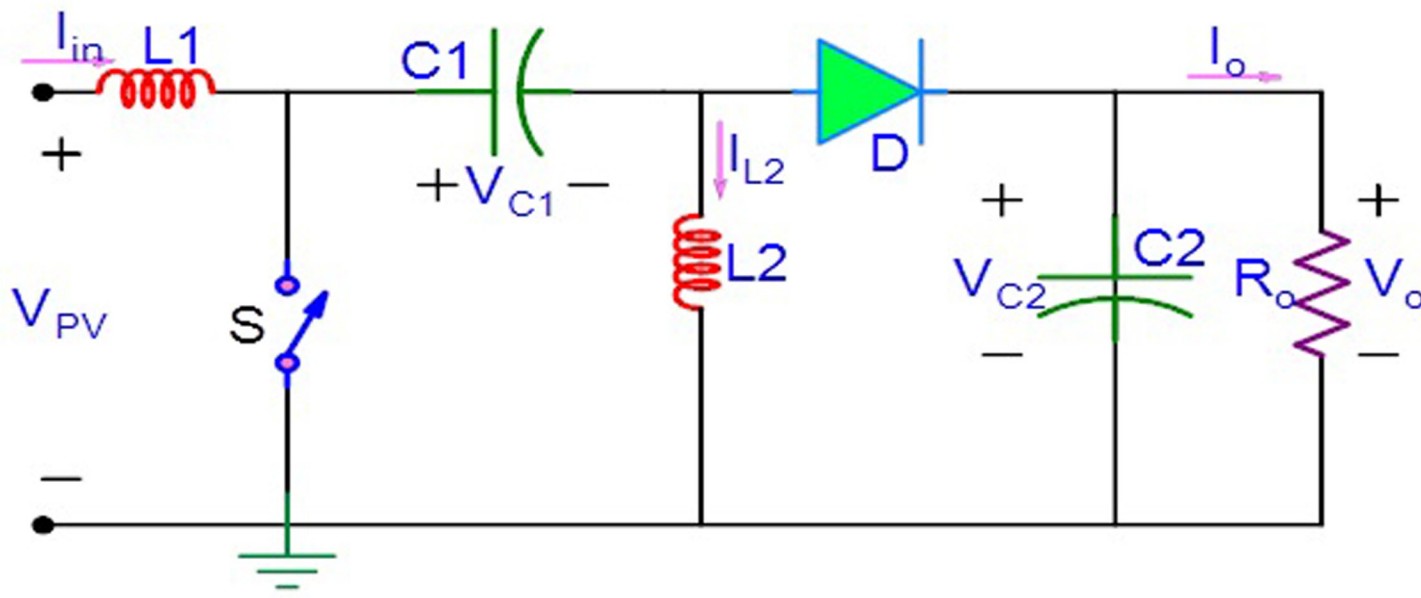

**Fig 5. Single ended primary inductance converter.**

$$C_1 \frac{dV_{C1}}{dt} = (1-k)I_{L1} - kI_{L2} \tag{7}$$

$$C_2 \frac{dV_{C_2}}{dt} = (1-k)(I_{L1} + I_{L2}) - \frac{V_{C2}}{R} \tag{8}$$

The mathematical correlations between input resistance observed by the converter and load resistance can be used to define MPPT capacity [29]. With the basic formulae of DC-DC converters, the input resistance of the converter, which corresponds to the effective resistance of the PV module, can be calculated [29].

The input voltage range of SEPIC is greater than that of buck and boost topologies. Because, if practical constraints are ignored, input voltage can be preferably zero or infinite [29]. This feature enables MPPT process to be carried out flawlessly under varying solar irradiation and load resistance situations [29]. The resistance of a PV module ($R_{PV}$) can be computed as follows.

$$V_O = I_O * R_O = \left(\frac{k}{1-k}\right) * V_{PV} \tag{9}$$

$$R_O = \left(\frac{k}{1-k}\right) * \frac{V_{PV}}{I_{PV}} * \left(\frac{k}{1-k}\right) \tag{10}$$

$$R_{PV} = \left(\frac{1-k}{k}\right)^2 * R_O \tag{11}$$

## 4. Arithmetic optimization algorithm (AOA)

At large, population-based optimization algorithms initiate their improvement procedures with a set of randomly produced solutions [52]. This created candidate solution is progressively enhanced by a set of rules and repeatedly assessed by an exact goal function; this is the essence of optimization methods [52]. Because population-based algorithms try to find the optimum result to optimization problems in a stochastic manner, obtaining a result in a sole run is not guaranteed [52]. Nonetheless, a numerous random solutions and optimization rounds enhances the likelihood of getting the global best solution for the given problem [53].

The optimization process is divided into two stages: exploration and exploitation, despite the variations among meta-heuristic algorithms used in population-based optimization methods [54]. In order to avoid local solutions, the process relates to thorough search space utilizing search agents of an algorithm. The latter is concerned with enhancing the accuracy of solutions found during the exploration phase [52]. Fig 6 depicts the clear and thorough AOA process.

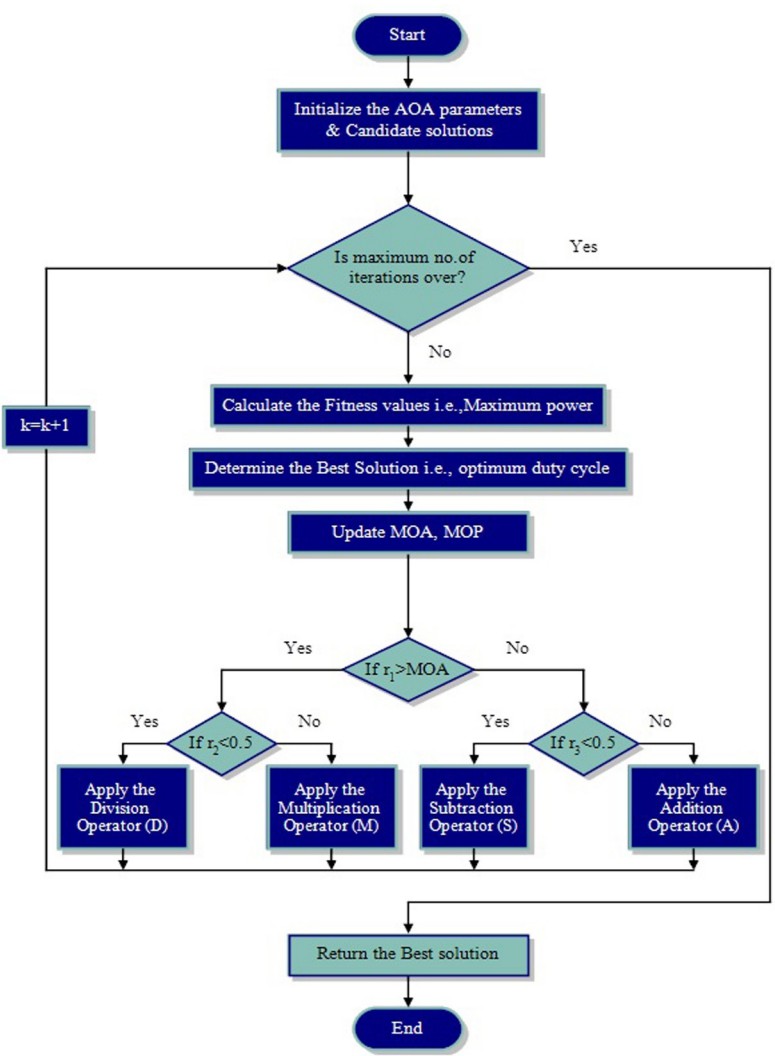

**Fig 6. Flowchart of arithmetic optimization algorithm for tracking MPP.**

The exploration and exploitation process have been denoted in the next sub-sections, which is attained by the Arithmetic operators i.e., Multiplication (M "×"), Division (D "÷"), Subtraction (S "−"), and Addition (A "+") [52, 53]. This meta-heuristic technique uses population data to solve optimization problems without figuring out their derivatives. Prior to starting its work, the AOA should choose a search method (i.e., exploration or exploitation). As a result, the Math Optimizer Accelerated (MOA) function is a coefficient defined by Eq 12 in the following search steps [53].

$MOA^k$ denotes the value at the $k^{th}$ iteration of the MOA function, as assessed by Eq 12. The current iteration is $k$: [1. . . . . . . . .N]. The accelerated function values (minimum and

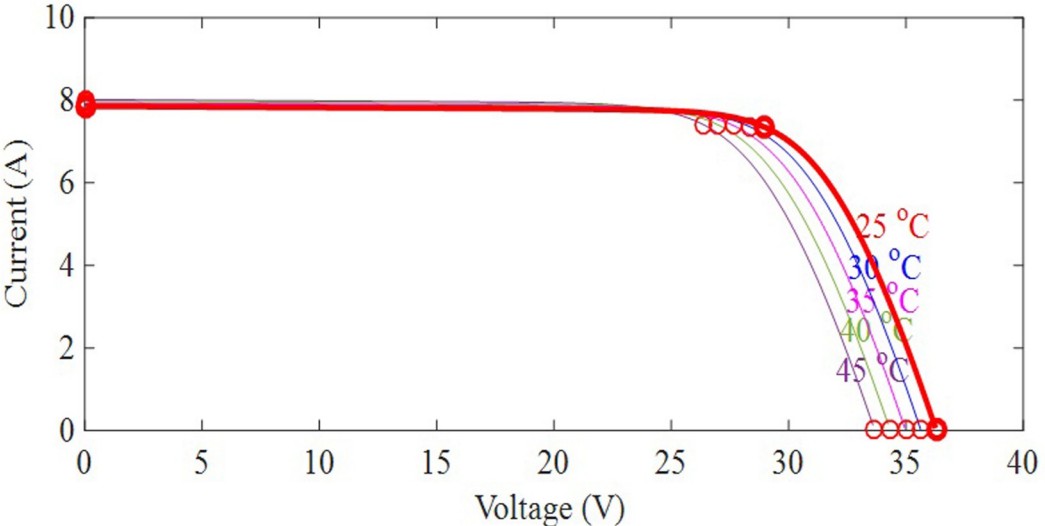

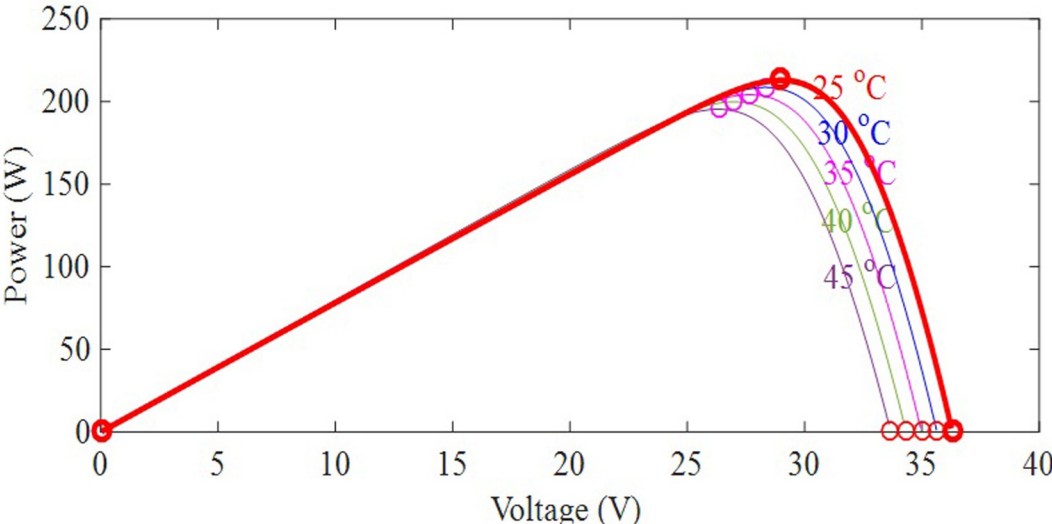

**Fig 7. Effect of temperature variations on PV module.**

maximum) are min and max, respectively.

$$\text{MOA}^k = \text{min} + k \times \left( \frac{\text{max} - \text{min}}{N} \right) \tag{12}$$

### 4.1 Exploration phase

Eq 13 represents the AOA exploration operators. The D or M operators are used in the exploration phase when $r_1 >$ MOA. D operator has been applied when $r_2 < 0.5$, alternatively, the M operator has been applied. $r_2$ is a chance number. The location updating procedure is represented by Eq 13.

$$x_{i,j}^{k+1} = \begin{cases} best\left(x_j\right) \div (MOP + \epsilon) \times \left((UB_j - LB_j) \times \mu + LB_j\right), r_2 < 0.5 \\ best\left(x_j\right) \times (MOP) \times \left((UB_j - LB_j) \times \mu + LB_j\right), otherwise \end{cases} \tag{13}$$

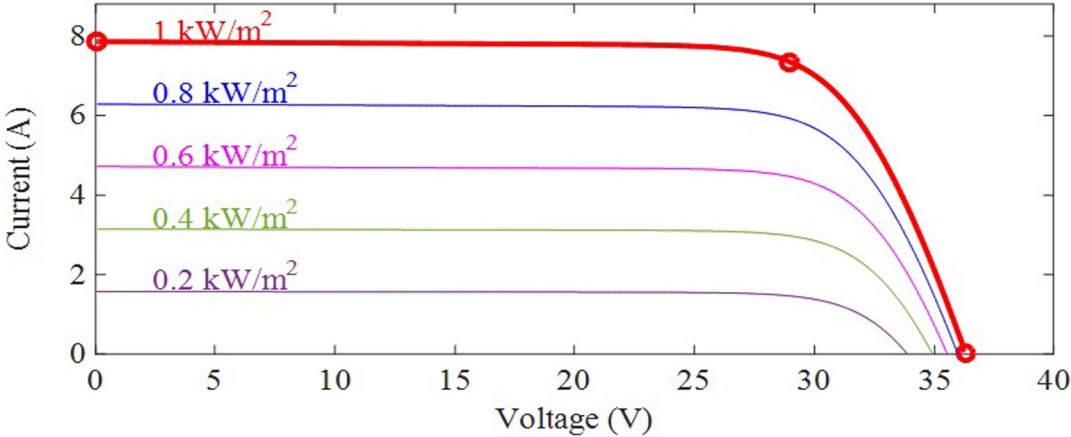

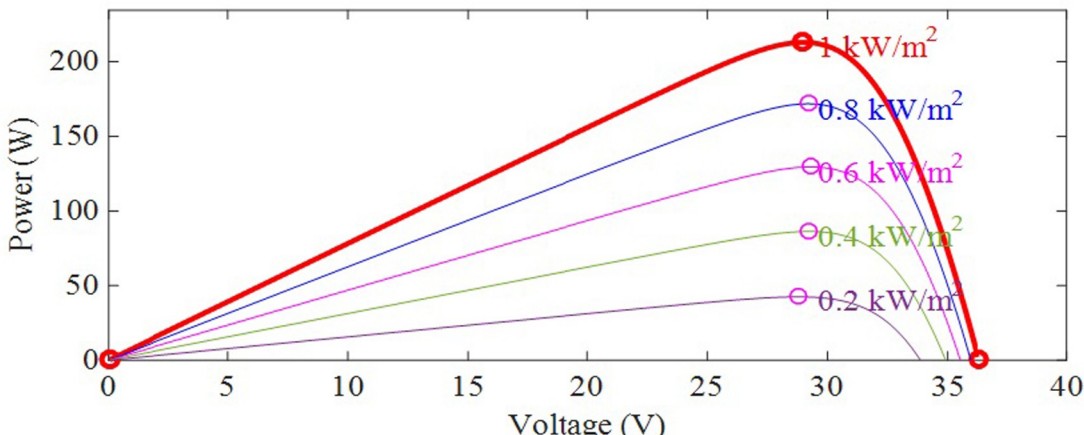

**Fig 8. Effect of insolation variations on PV module.**

where

$$MOP^k = 1 - \frac{k^{(1/\alpha)}}{k_{max}}$$

Math Optimizer probability (MOP) is a coefficient. The two parameters are $\mu$ and $\alpha$. The results of this research indicate that whereas $\alpha$ is a sensitive parameter that specifies the exploitation accuracy across the iterations and is fixed at 5, $\mu$ is a control parameter that adjusts the search process and is fixed at 0.5. $\epsilon$ is a random number between 0 and 1. Upper bound and lower bound values represented by UB and LB, respectively.

## 4.2 Exploitation phase

During the exploitation phase, the S and A operators are trained by the MOA function value. Eq 14 represents search strategies for S and A.

$$x_{i,j}^{k+1} = \begin{cases} best\left(x_j\right) - (MOP + \epsilon) \times \left(\left(UB_j - LB_j\right) \times \mu + LB_j\right), r_3 < 0.5 \\ best\left(x_j\right) + (MOP) \times \left(\left(UB_j - LB_j\right) \times \mu + LB_j\right), otherwise \end{cases} \tag{14}$$

AOA tunes the duty cycle for SEPIC by calculating fitness function at every iteration. The best value of the duty cycle for each generation has been calculated after assessing the objective function. In this paper, the objective of optimization problem is power maximization. The constraints are that duty cycle must lie between 0 and 1.

## 5. Results and discussion

The MPPT capability of a solar PV is obtained by the DC-DC converter, solar irradiation, temperature, and load resistance value. When all of these characteristics are considered in an MPPT application, theoretical instant tracking efficiency may be determined as shown below.

$$\eta = \frac{P(G, T_C, R_L)}{P_M} \tag{15}$$

These three parameters $(G, T_C, R_L)$ have different effects on tracking efficiency for MPPT capabilities. For example, fluctuations in solar irradiation and environmental temperature affect the junction temperature of a PV module. In other words, high sun irradiation causes high temperatures. As a result, increasing the temperature significantly reduces the open circuit voltage in compared to the short circuit current, as illustrated in Fig 7.

The load resistance is the second parameter. MPPT capability reduces if the load resistance value is not suitable. Meanwhile, the variations of solar insolation have a considerable impact on the PV module's I-V curve, as illustrated in Fig 8. Because the temperature influence is negligible in comparison to solar irradiation, it is ignored in this study.

**Table 1. Specifications of PV module.**

| Parameters | Values |
|---|---|
| Short circuit current ($I_{sc}$) | 8.83A |
| Open circuit voltage ($V_{oc}$) | 36.8V |
| Current at MPP ($I_{mpp}$) | 8.3A |
| Voltage at MPP ($V_{mpp}$) | 30V |
| Maximum power ($P_{max}$) | 249W |

**Table 2. SEPIC parameters.**

| Parameters | Values |
|---|---|
| Switching frequency, fs | 50KHz |
| Inductor 1, L1 | 1.1478e-3H |
| Inductor 2, L2 | 1.478e-3H |
| Capacitor 1, C1 | 0.5e-3F |
| Capacitor 2, C2 | 0.5e-3F |
| Resistive load, R | 40Ω |

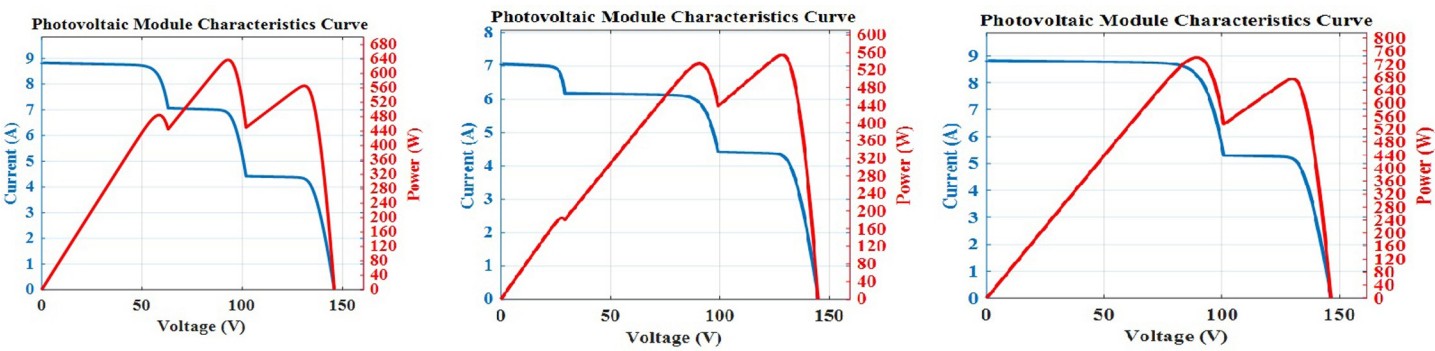

**Fig 9. PV characteristics under NUCs with different GP positions GMPP at middle, GMPP at end, GMPP at start.**

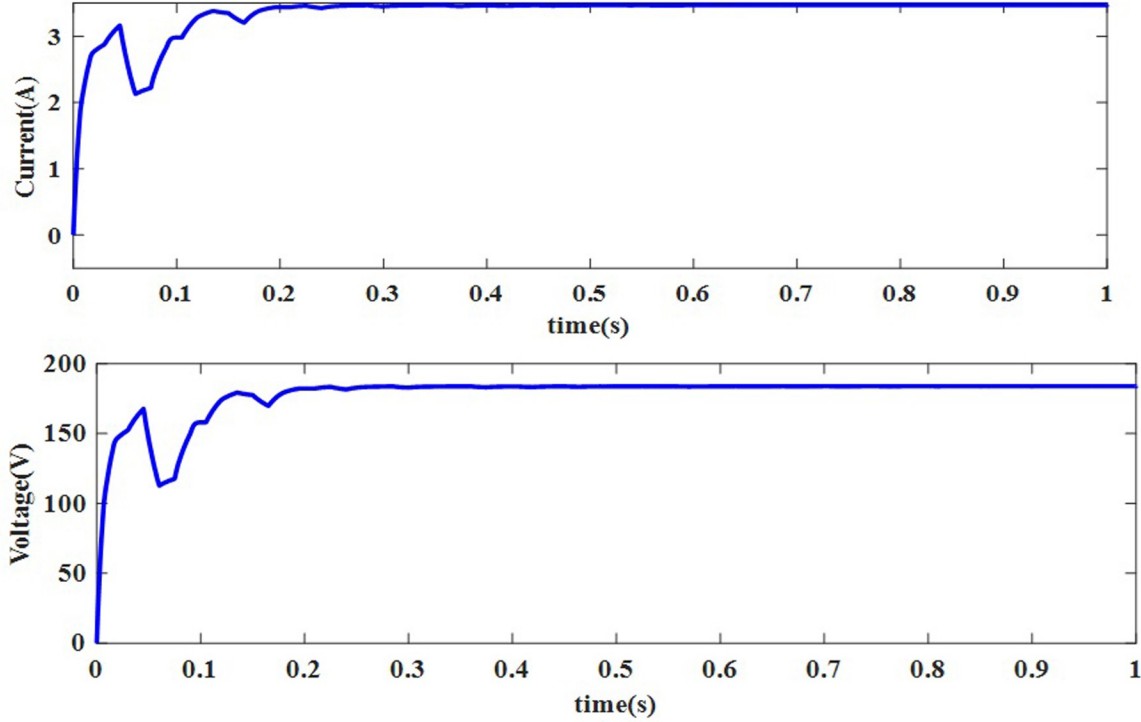

**Fig 10. Output across the load current and voltage.**

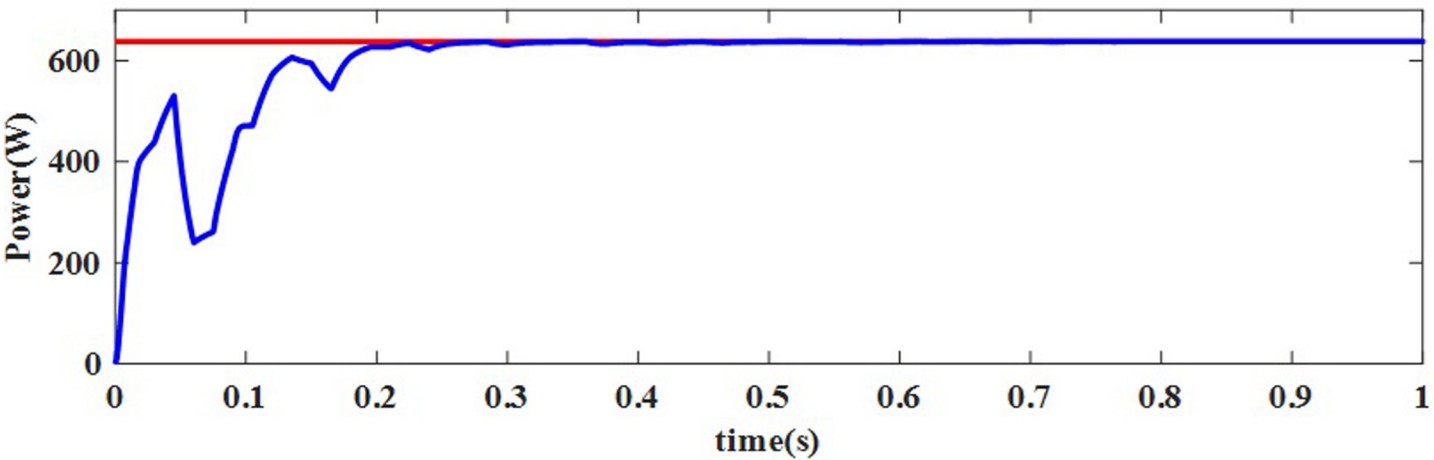

**Fig 11. Maximum power delivered to the load for NUC-1.**

MPPT is achieved regardless of load resistance or sun irradiation. Load curves begin when $R_{PV}$ is zero and conclude when $R_{PV}$ is infinite. In other words, there are no limits linked to solar irradiation and load resistance for all load circumstances. Theoretical load resistance ranges from 0 to $\infty$. As a result, SEPIC is regarded to be the optimum solution for MPPT applications. Table 1 shows the electrical properties of the PV module utilized in the study, which is the "Tata Solar PV system".

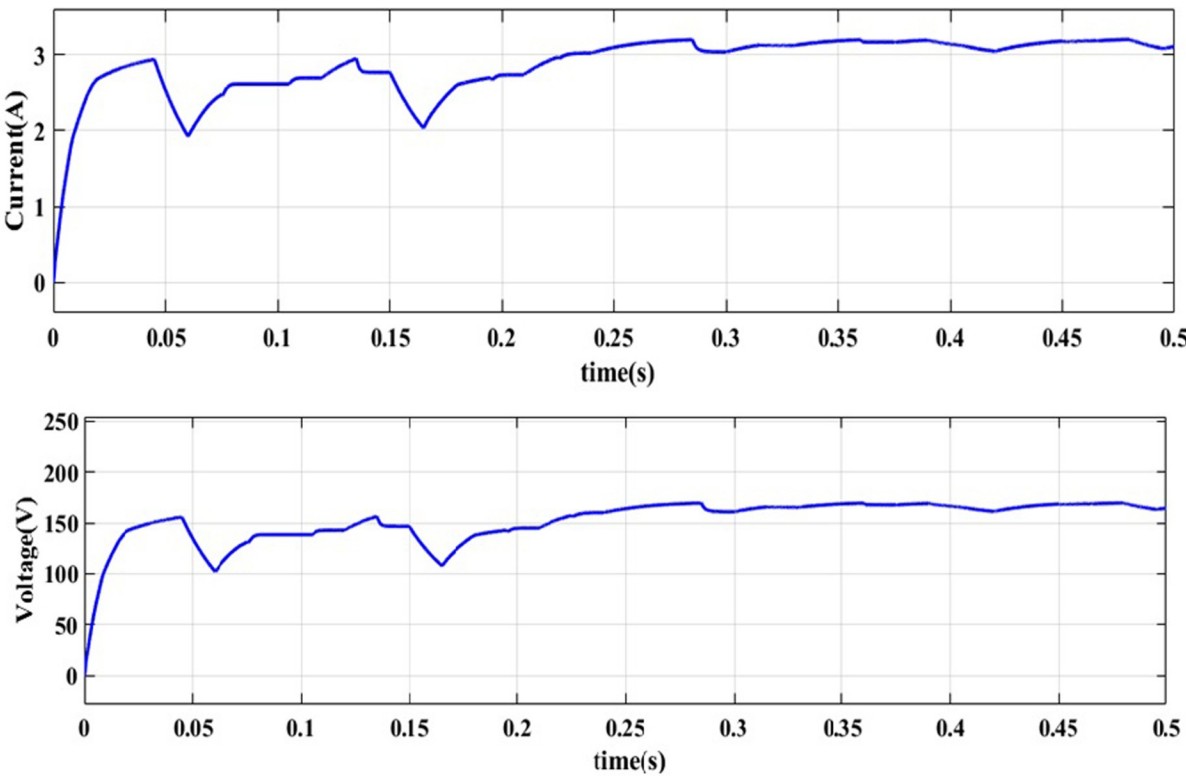

**Fig 12. Output obtained for NUC-2 current and voltage.**

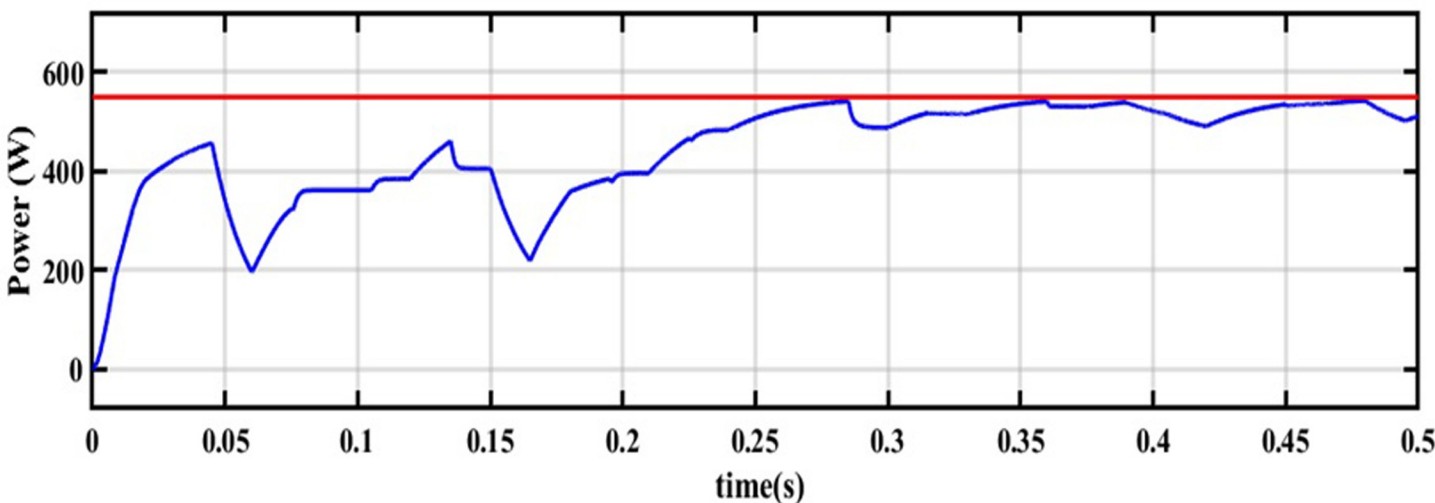

**Fig 13. Output power delivered to the load under case 2 conditions.**

The point of maximum power varies with temperature and sun irradiation. When this happens, the MPP must be traced by changing the array terminal voltage using the SEPIC converter. Table 2 specifies the SEPIC converter's parameters.

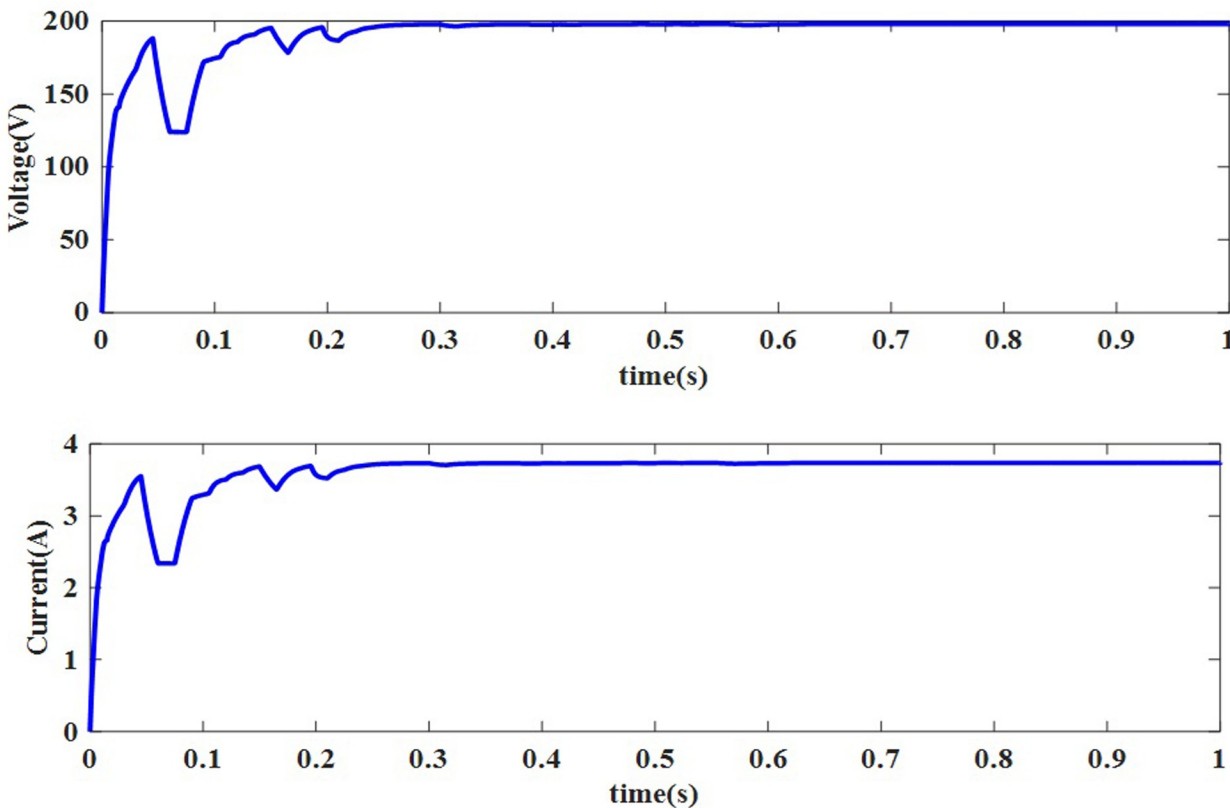

**Fig 14. Output obtained for NUC-3 current and voltage.**

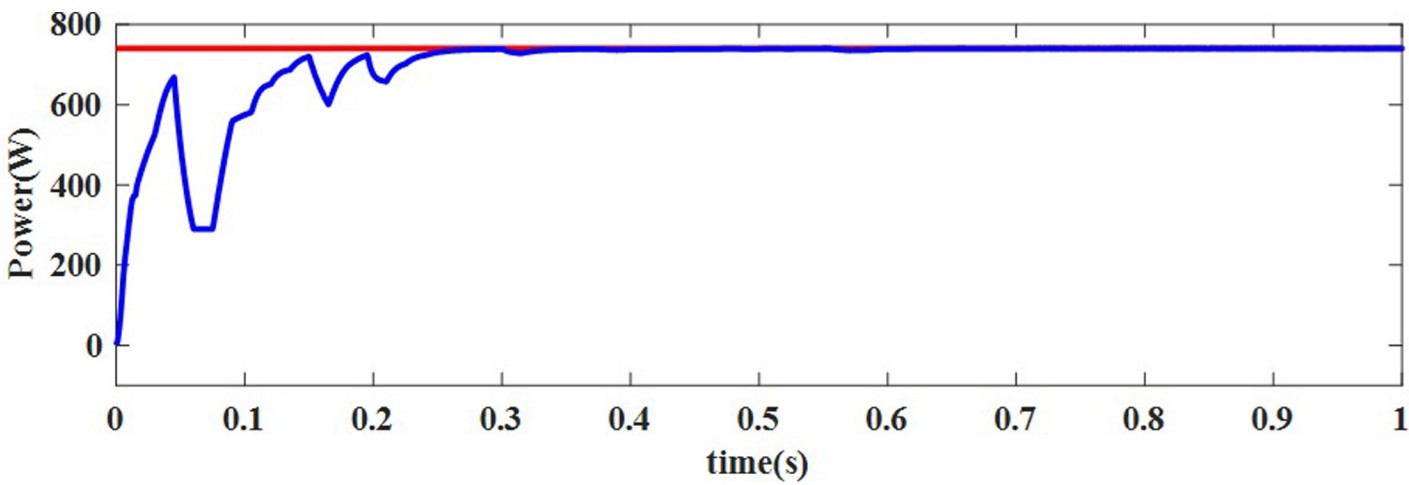

**Fig 15. Output power delivered to the load under case 3 conditions.**

AOA based MPPT algorithm has been validated under both partial shaded and rapid changing insolation situations. Under partial shaded conditions, three different nonuniform conditions (NUC) are used for MPPT algorithm verification.

### 5.1 Nonuniform environments

A computer simulation of a solar PV system is run at a constant temperature of 25˚C with various PV module insolation levels. The proposed AOA-MPPT's efficacy under different insolation patterns has been validated using three nonuniform conditions in terms of convergence, tracking speed, steady state oscillations, and tracking efficiency. The tracking efficiency ($\eta$) is calculated by dividing the average output power obtained under steady-state conditions by the maximum attainable power of the PV array under a specific pattern [55, 56]. Three NUCs have been chosen in which GMPP is located in middle, end and start positions, as shown in Fig 9 respectively.

**NUC-1:** Insolation = [600 800 700 900] $W/m^2$ and Temperature = 25˚C.

For this NUC-1, the output voltage, current and power of solar PV array for insolation of [600 800 700 900] $W/m^2$ have nonlinear responses and a fluctuating output voltage with a high ripple, which is regulated by a SEPIC converter. The output waveform of the SEPIC is depicted in Fig 11.

In Fig 11, the actual MPP represented as constant whereas the AOA-MPPT tracks the actual MPP curve in 0.2 seconds. Figs 10 and 11 indicate that the proposed MPPT tracks the GP faster and closely matches the actual MPP. From Fig 11, the convergence time with AOA is

**Table 3. Numerical results obtained from AOA and GWO method.**

| Algorithm | Condition | MAE | MSE | RMSE | Mean | SD | SD in steady state |
|---|---|---|---|---|---|---|---|
| AOA | NUC1 | 34.5 | 8.9031e+03 | 94.35 | 603.2080 | 0.0459 | 0.0029 |
| | NUC2 | 108.0709 | 2.1478e+04 | 146.55 | 441.9291 | 0.1368 | 0.0316 |
| | NUC3 | 28.7572 | 7.5173e+03 | 86.7026 | 711.6504 | 0.0310 | 0.0019 |
| GWO | NUC1 | 45.6 | 9.686e+03 | 98.42 | 599.3480 | 0.052 | 0.0042 |
| | NUC2 | 120.34 | 2.3311e+04 | 152.68 | 420.5824 | 0.142 | 0.0426 |
| | NUC3 | 33.26 | 8.619e+03 | 92.84 | 700.7824 | 0.036 | 0.0026 |

**Table 4. Results of PV array under various insolation levels.**

| Case study | Insolation levels (W/m²) | | | | Maximum Power (W) | | | | Settling time (s) | | | Tracking Efficiency (%) | | |
|---|---|---|---|---|---|---|---|---|---|---|---|---|---|---|
| | | | | | Actual | PO | GWO | AOA | PO | GWO | AOA | PO | GWO | AOA |
| Case 1 | 600 | 800 | 700 | 900 | 640.6 | 240 | 625 | 637 | 0.15 | 0.35 | 0.2 | 37.46 | 97.56 | 99.44 |
| Case 2 | 500 | 800 | 700 | 700 | 550.4 | 400 | 548.3 | 550 | 0.2 | 0.42 | 0.28 | 72.67 | 99.62 | 99.93 |
| Case 3 | 1000 | 600 | 1000 | 1000 | 740.4 | 720 | 738 | 740 | 0.25 | 0.4 | 0.25 | 97.24 | 99.67 | 99.94 |

200ms for non-uniform condition-1. According to Table 4, the suggested MPPT has faster convergence than the GWO and PO MPPTs. The tracking efficiency of the PO-MPPT, GWO-MPPT and AOA-MPPT is 37.63%, 97.56% and 99.44%, respectively. Even though both optimization algorithms give near optimal results, the settling time of the AOA-MPPT is reduced by 42.85% as compared to the GWO-MPPT. As a result of the findings, it can be concluded that the proposed method gives superior performance in terms of quicker convergence to the GP and higher energy productivity under a variety of shifting insolation patterns.

**NUC-2:** Insolation = [500 800 700 700] $W/m^2$ and Temperature = 25°C

A computer simulation of a solar PV system is carried out at a constant temperature of 25°C with different insolation levels i.e., [500 800 700 700] $W/m^2$ of PV modules in PV array. To track GP, the suggested AOA-MPPT procedure is applied to the same 4S arrangement for NUC-2. The suggested system's key waveforms for this operating situation are depicted in Figs 12 and 13. For this NUC-2, the proposed algorithm takes 0.28 seconds whereas the GWO algorithm takes 0.42 seconds to track MPP. To track the MPP, the duty cycle varies with changes in insolation of PV modules. The suggested AOA-MPPT has reduced the fluctuations near MPP and enhanced the steady-state performance, as shown in the Fig 13. It's also worth noting that the proposed technology offers a quick transient reaction. When the solar irradiation level of all PV modules in PV array is nonuniform, the operational point of the arrangement changes extremely quickly to monitor the MPP of the PV array without fluctuation.

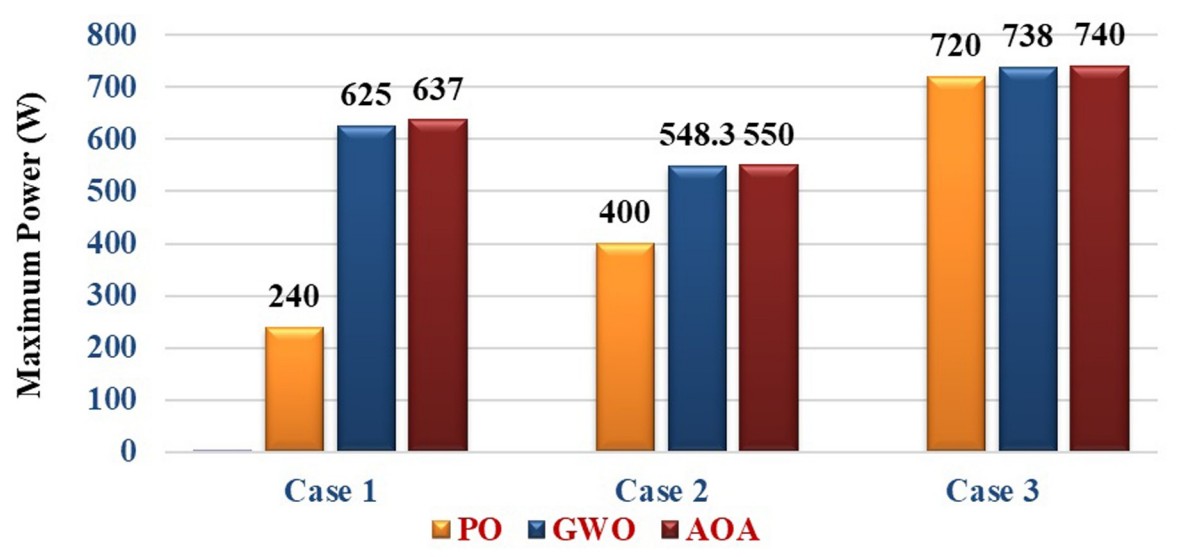

**Fig 16. Maximum power obtained by three MPPT methods.**

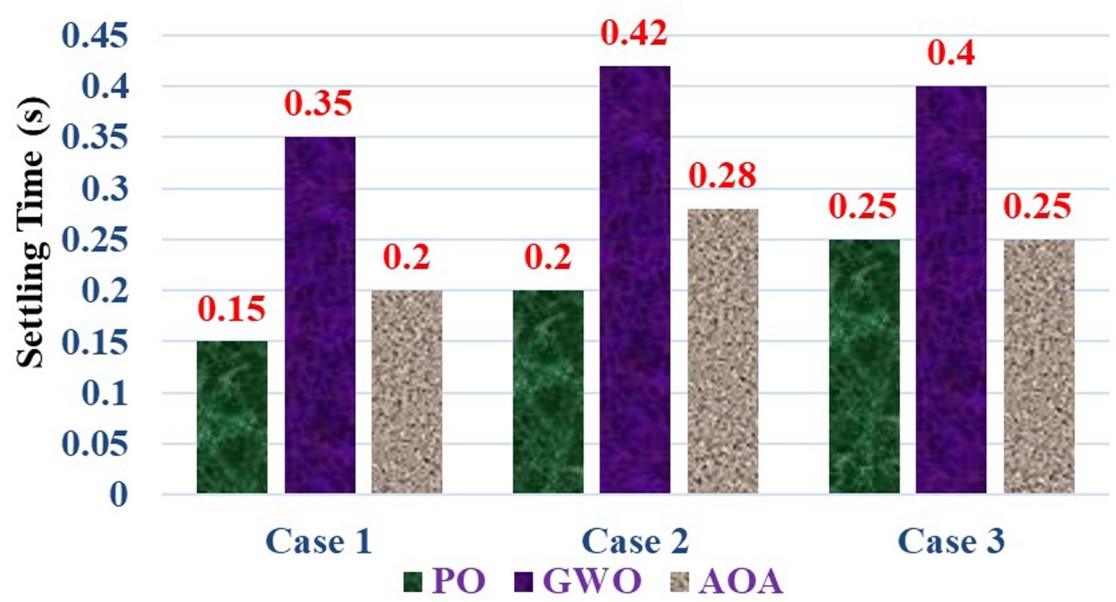

**Fig 17. Evaluation of three MPPT approaches in terms of settling time.**

**NUC-3:** Insolation = [1000 600 1000 1000] $W/m^2$ and Temperature = 25°C

A computer simulation of a solar PV system is carried out at a constant temperature of 25°C with different insolation levels i.e., [1000 600 1000 1000] $W/m^2$ of PV modules in PV array. To track GP, the suggested AOA-MPPT procedure is applied to the same 4S arrangement for NUC-3. The output waveforms are shown in Figs 14 and 15. The GMPP has been tracked in 0.25s with the tracking efficiency of 99.94%.

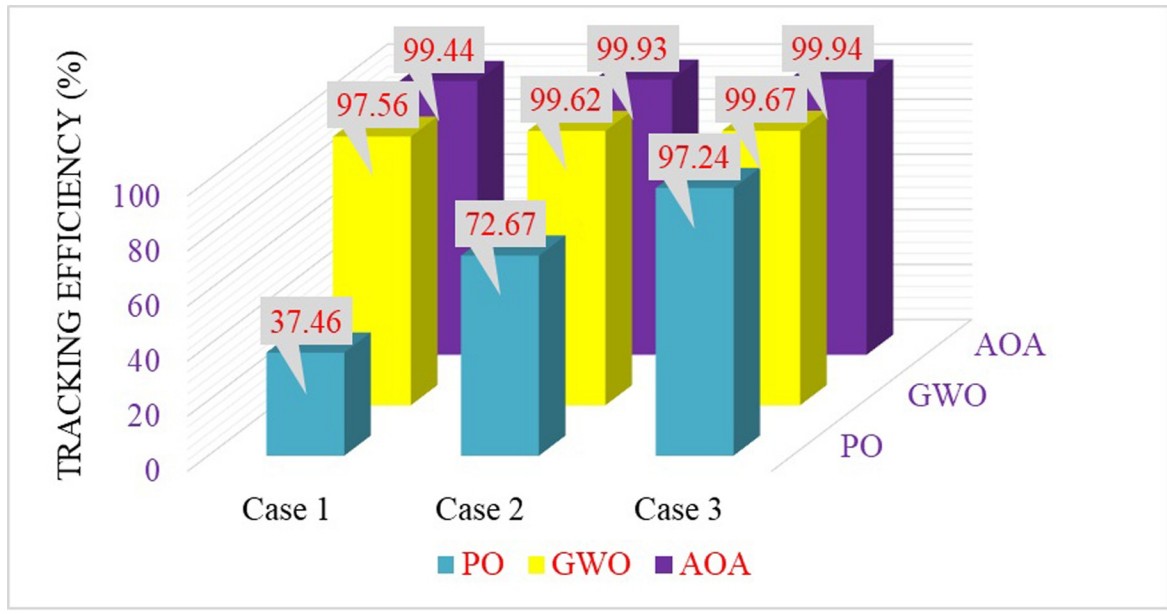

**Fig 18. Comparison of the maximum power point tracking efficiency.**

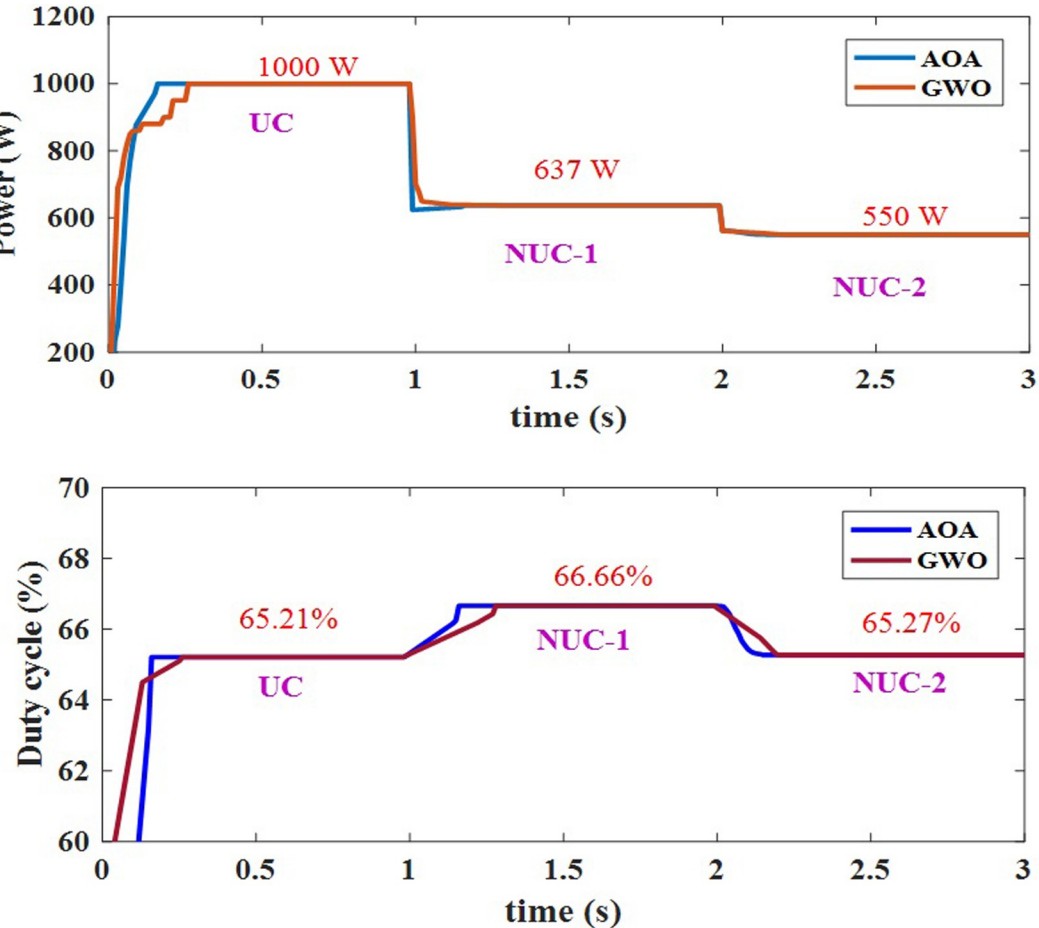

**Fig 19. Maximum power obtained and duty cycle during case 1 of fast changing irradiance conditions with GWO-MPPT 280ms and GWO-MPPT 120ms.**

The statistical terms such as, mean, standard deviation (SD), mean absolute error (MAE), mean square error (MSE), and root mean square error (RMSE) are used to analyse the results under non uniform conditions, as illustrated in Table 3. In addition, standard deviation of power tracked after the MPP is settled (in the steady-state) is tabulated to quantify the amount of oscillations. The results of the PV array during three cases obtained by PO, GWO and AOA have been compared and it is given in Table 4.

According to Table 4, the PO algorithm tracks the MPP and converges to the first peak point. But the GWO and AOA methods have been efficiently tracks the Global MPP with a short time. The maximum power obtained, settling time, and Tracking efficiency of three MPPT approaches are compared and depicted in Figs 16–18. According to the simulation results, it is concluded that the AOA-MPPT outperforms in terms of faster convergence (low settling time) to GP, higher efficiency, and less oscillations.

## 5.2 Fast changing irradiance conditions

The solar intensities of the PV modules are arbitrarily adjusted and the 4s configuration has been exposed to fast changing irradiance conditions i.e., UC switches to NUC-1 and then NUC-2. Each condition is in possession for 1 second. Two different cases for fast changing

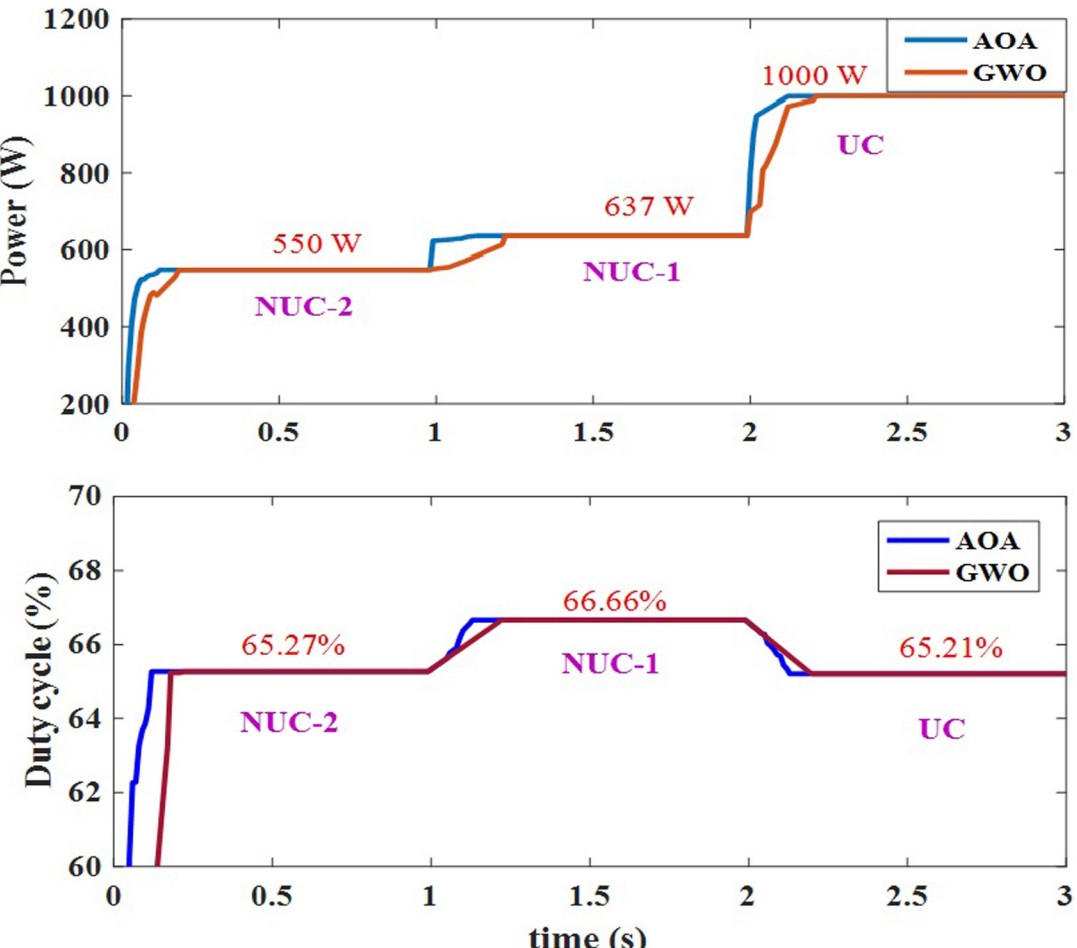

**Fig 20. Tracking curves of AOA and GWO algorithm for case 2 of fast changing irradiance conditions with power and duty cycle.**

irradiance conditions have been considered to prove the dynamic behaviour of the AOA-MPPT. In the first case, each pattern such as UC, NUC-1 and NUC-2 are exposed at 1 second each. The extreme power obtained from PV array and the corresponding duty cycle variations for each pattern are depicted in Fig 18. According to this, the convergence time with GWO-MPPT for UC is 280ms and it can be reduced to 160ms with the aid of AOA-MPPT. And convergence time is reduced to 120ms with AOA and 150ms with GWO for change in irradiance condition from UC to NUC1.

In the second case, the NUC-2 switches to NUC-1 and then switches to UC. The tracking curves of the case 2 during NUC-2, NUC-1 and UC are shown in Fig 19. The examples above

**Table 5. Comparative performance of the suggested MPPT with the other MPPTs.**

| MPPT method | Precision | Speediness | Convergence rate | Steady state Fluctuations | Efficacy |
|---|---|---|---|---|---|
| PO-MPPT | Less | Slow | Possible to occur in LP | Occurred | Low |
| GWO-MPPT | Moderate | Moderate | High | Not occurred | Moderate |
| AOA-MPPT | More | Rapid | High | Not occurred | High |

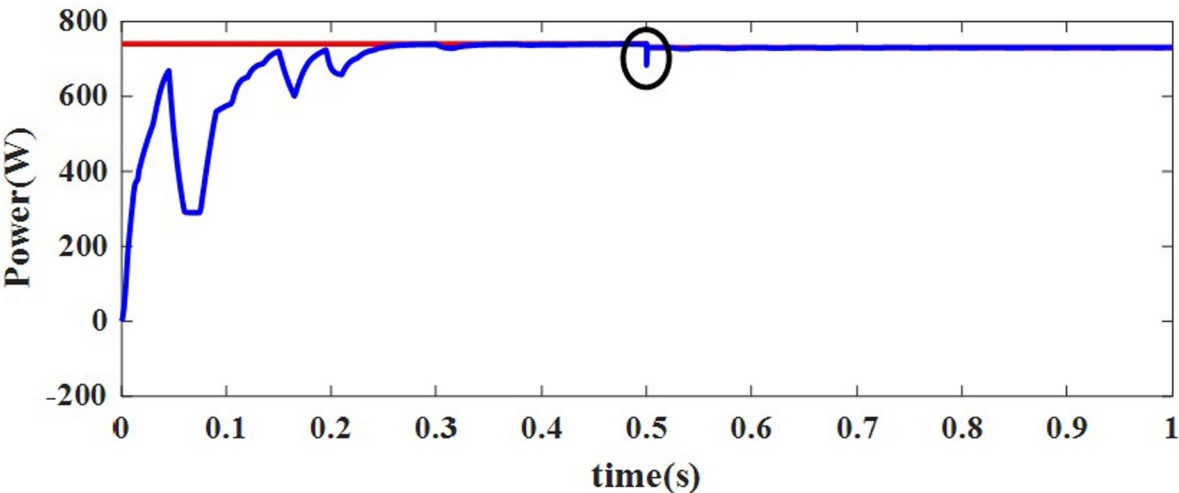

**Fig 21. Output power due to load variations.**

demonstrate that the AOA-MPPT and GWO-MPPT can follow GP with no oscillations. But the tracking speed of the proposed AOA-MPPT superior than the GWO-MPPT.

Fig 20 show that the proposed AOA-MPPT converges to the GP quicker than the GWO-MPPT. Many peaks with various local peaks and one global peak characterise the P–V curve under nonuniform insolation conditions. It is worth noting that when the mathematical operators find the MPP, the duty cycle is maintained at a constant value, which eliminates the steady-state oscillations that can occur with traditional MPPT techniques. These graphs show how the AOA-MPPT algorithm can track the maximum power point and transfer power from the PV module to the load resistance. The SEPIC converter has been developed to give good regulation over rapid voltage fluctuations with minimal ripple. The performance comparison of PO, GWO and the proposed AOA-MPPT method is given in Table 5.

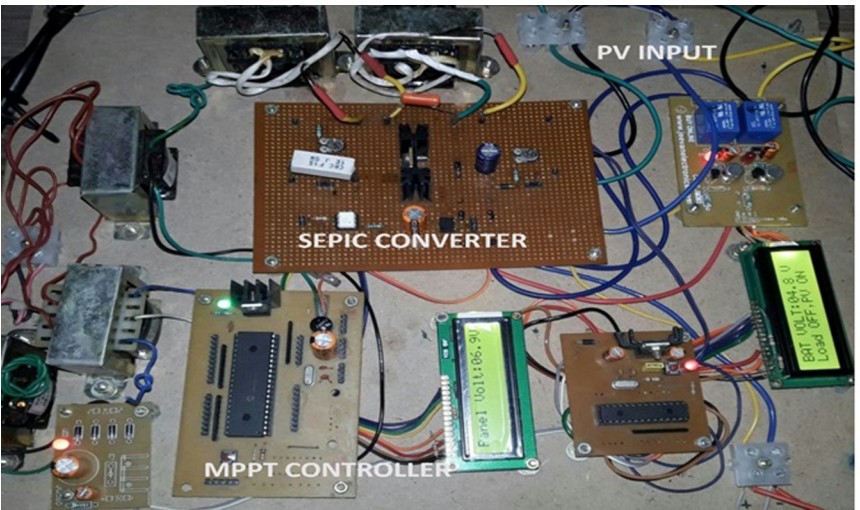

**Fig 22. Experimental setup for the proposed PV system.**

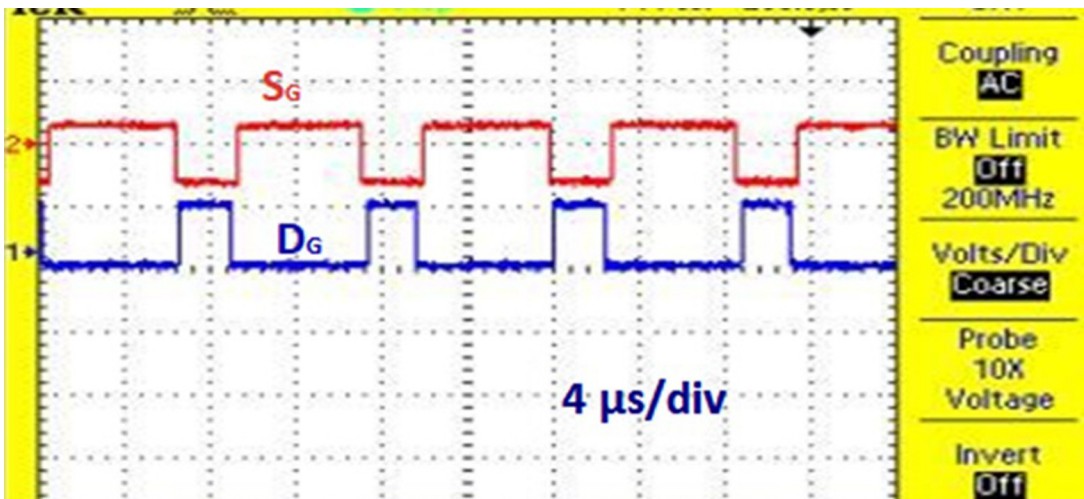

**Fig 23. Experimental results for switching pulses.**

## 5.3 Validation of proposed method under load variations

The insolation values are set at [1000 600 1000 1000] W/m$^2$ for the GMPP at the start position. The GMPP has been tracked at 250ms. While temperature is constant, The PV power can change with the insolation changes or load variations. When the load variation occurs once the MPP is tracked, the PV current is varied. By adjusting duty cycle, the output power quickly tracks actual power. Fig 21 is included to describe the effect of load variation on proposed method. When resistive load has been reduced from 40Ω to 20Ω at 0.5seconds, the disturbance occurs in output voltage of converter but the AOA tracks the actual power within 10ms.

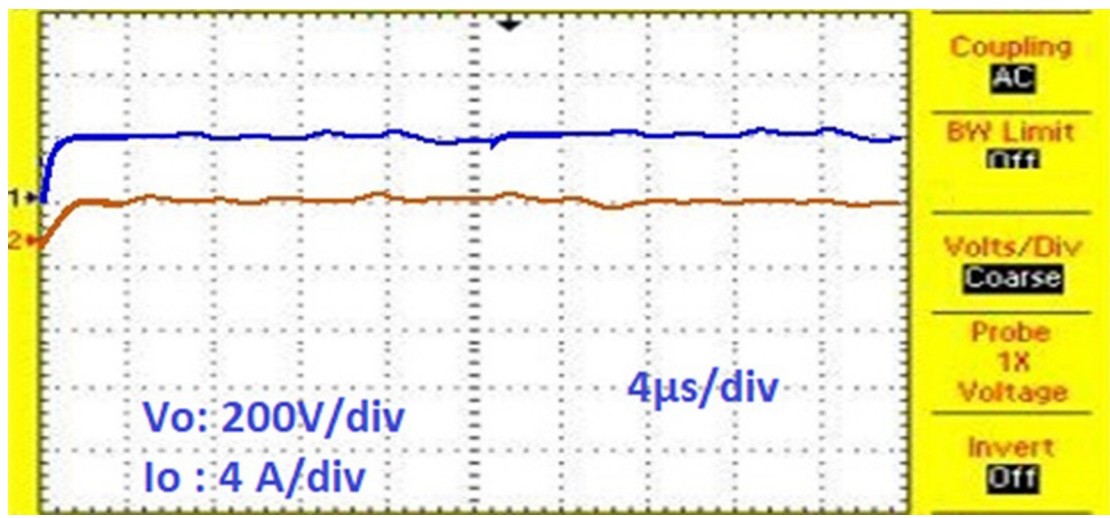

**Fig 24. Output voltage and current of converter.**

## 6. Experimental results

The experimental setup for the proposed system is developed according to the simulation specification as shown in Fig 22. The proposed control algorithm for the DC-DC converter is implemented using PIC 16F877A from Microchip.

Fig 22 shows the experimental results for proposed solar PV system under UC where the results obtained are similar to the results obtained from simulation. Fig 23 shows the control signal obtained from the proposed AOA-MPPT algorithm to the switches which also shows the voltage across the diode. Fig 24 shows the converter's steady state output voltage of 198 V and current of 3.74 A when **NUC-3:** Insolation = [1000 600 1000 1000] $W/m^2$ and Temperature = 25˚C. Fig 25 depicts the variation of voltage and current during non-uniform environmental conditions described in NUC-3. Fig 26. depicts the variation of voltage and current during fast changing solar irradiance condition. The experimental results evidences the efficacy of proposed AOA-algorithm under different circumstances and closely matches with simulation results.

## 7. Conclusion

This paper presents the design of an Arithmetic optimization algorithm (AOA)-based maximum power point tracker that can act according to varying insolation levels of PV modules. The PV array's output power varies greatly depending on solar insolation and temperature. The PV array is operated at its maximum operational point, where the supreme power generated may be transferred to the load connected across the SEPIC converter's output terminal, using MPPT control method. The AOA is a recently developed optimization that overcomes issues like low tracking efficiency and steady-state fluctuations. The AOA-based maximum power point tracker delivers greater MPP chasing and has a quicker convergence compared than perturb & observe algorithm and grey wolf optimization algorithm. From the results, the suggested AOA-MPPT gets tracking efficiency of above 99% and settling time of 200 to 300ms under non uniform conditions. When the operating point converges to MPP, the system can retain maximum power while keeping the duty cycle constant. The efficacy of the proposed MPPT method is confirmed for both partial shaded conditions and rapid changing irradiance conditions. Comparative results show that the proposed method exhibits greater performance

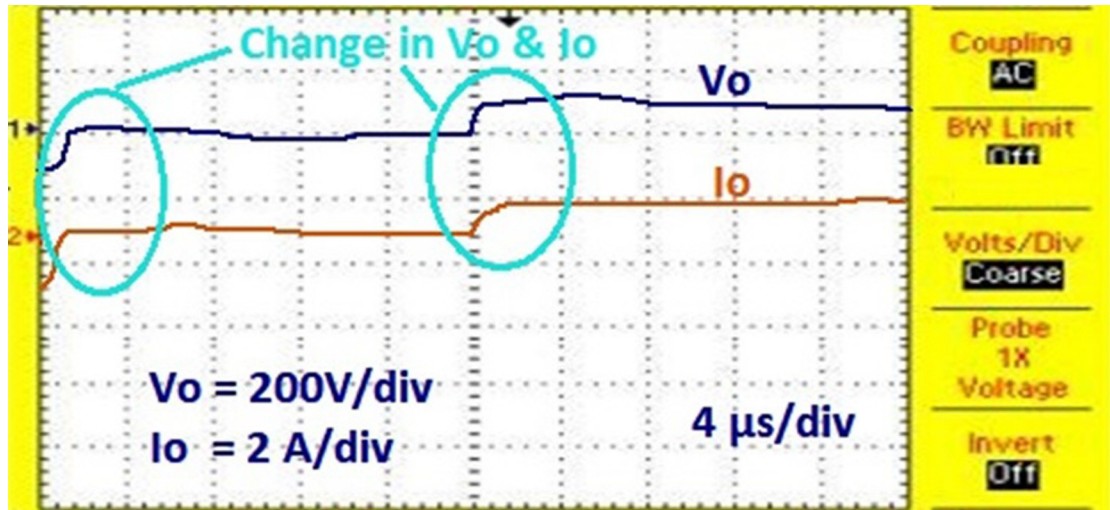

**Fig 25. Variation of voltage and current during non-uniform environmental conditions.**

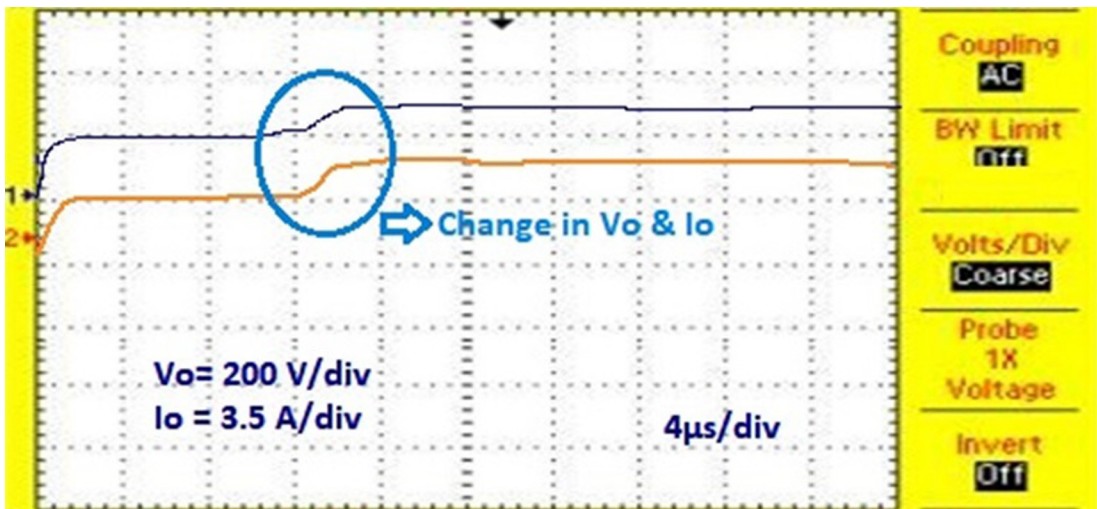

**Fig 26. Variation of voltage and current during fast changing solar irradiance conditions.**

than other methods under steady state and dynamic conditions. In light of these findings, the work's next stage will involve hardware validation of the suggested AOA-MPPT algorithm for PV systems operating in uneven and quickly changing settings. It is anticipated that the PV community, comprising researchers and practitioners alike, will be highly interested in our work. In order to build a solar photovoltaic grid-connected power generating system in the future, authors will incorporate the suggested AOA-MPPT technique with PV inverters, with the goal of enhancing the overall energy harvesting efficiency. Analysis can be done on any single switch DC-DC boost converter with numerous peaks in the P-V curve can use the AOA-MPPT. In addition, the authors will take into account a practical solution to address the issue of partially shadowing the PV array surface.

## Author Contributions

**Conceptualization:** Karthikeyan Balasubramani, Sundararaju Karuppannan.

**Data curation:** Karthikeyan Balasubramani, Sundararaju Karuppannan.

**Formal analysis:** Sundararaju Karuppannan.

**Funding acquisition:** Zakaria M. S. Elbarbary, Saad F. Al-Gahtani.

**Investigation:** Zakaria M. S. Elbarbary.

**Methodology:** Durga Devi Ravichandran.

**Project administration:** Zakaria M. S. Elbarbary, Saad F. Al-Gahtani.

**Resources:** Nageswari Sathiyamoorthy, Durga Devi Ravichandran, Karthikeyan Balasubramani.

**Software:** Durga Devi Ravichandran, Karthikeyan Balasubramani.

**Supervision:** Durga Devi Ravichandran, Zakaria M. S. Elbarbary, Ahmed I. Omar.

**Validation:** Maheshwari Adaikkappan, Nageswari Sathiyamoorthy, Zakaria M. S. Elbarbary, Ahmed I. Omar.

**Visualization:** Maheshwari Adaikkappan, Nageswari Sathiyamoorthy,
Zakaria M. S. Elbarbary, Saad F. Al-Gahtani, Ahmed I. Omar.

**Writing – original draft:** Maheshwari Adaikkappan, Nageswari Sathiyamoorthy,
Ramasamy Palanisamy.

**Writing – review & editing:** Maheshwari Adaikkappan, Nageswari Sathiyamoorthy,
Ramasamy Palanisamy, Saad F. Al-Gahtani.

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
