## [Decision Letter · Decision Letter 0]

25 Jun 2024

PONE-D-24-06244Arithmetic Optimization based MPPT for photovoltaic systems operating under non-uniform situationsPLOS ONE

Dear Dr. Omar,

Thank you for submitting your manuscript to PLOS ONE. After careful consideration, we feel that it has merit but does not fully meet PLOS ONE’s publication criteria as it currently stands. Therefore, we invite you to submit a revised version of the manuscript that addresses the points raised during the review process.

**ACADEMIC EDITOR:** The reviewers recommend reconsideration the manuscript with revision and modification. I invite the authors to resubmit the manuscript after addressing the comments raised by the reviewers.

We look forward to receiving your revised manuscript.

Kind regards,

Dhanamjayulu C, Ph.D & Post.Doc

Academic Editor

PLOS ONE

“the authors extend their appreciation to the Deanship of Scientific Research at King Khalid University under for funding this work through General Research Project under Grant number (RGP2/425/44).”

5. Please upload a copy of Figures, 1, 2,3, 4, 5, 6, 7, 8, 9, 10, 11, 12, 13, 14, and 15 to which you refer in your text on pages 4, 5, 6, 9, 10, 11, 12, 13 and 14. If the figure is no longer to be included as part of the submission please remove all reference to it within the text.

Additional Editor Comments:

The reviewers recommend reconsideration the manuscript with revision and modification. I invite the authors to resubmit the manuscript after addressing the comments raised by the reviewers.

Reviewers' comments:

Reviewer's Responses to Questions

**Comments to the Author**

1. Is the manuscript technically sound, and do the data support the conclusions?

Reviewer #1: Partly

Reviewer #2: No

Reviewer #3: Partly

Reviewer #4: No

2. Has the statistical analysis been performed appropriately and rigorously? 

Reviewer #1: No

Reviewer #2: No

Reviewer #3: No

Reviewer #4: I Don't Know

3. Have the authors made all data underlying the findings in their manuscript fully available?

Reviewer #1: No

Reviewer #2: No

Reviewer #3: Yes

Reviewer #4: No

4. Is the manuscript presented in an intelligible fashion and written in standard English?

Reviewer #1: Yes

Reviewer #2: Yes

Reviewer #3: Yes

Reviewer #4: No

5. Review Comments to the Author

Reviewer #1: - For some reasons, Figures are not included in the final PDF file. So, I am unable to comment on results.

- The work is limited to simulation. Mention, to which extent the designed optimization scheme is deployable on a real PV system? What are the challenges involved when the control law is to be implemented on a real system?

- Update the literature review by including notable works on MPPT control such as 10.3390/en16135039, 10.1371/journal.pone.0260480 and 10.3390/app12062773.

- Please write quantitatively especially in the Abstract: How much 'quick' is quick response? How much 'excellent' is excellent?

- Include more rigorous analysis based on ITAE, IAE, ISE etc.

- Explicitly mention novelty of the proposed optimization technique.

- Avoid referring to waveforms/curves in a graph from their color. e.g. The 'green dotted line outlines the ideal ..."

Reviewer #2: 1. Abstract is ambiguous, kindly arrange the abstract which can contain background of the work, objective, method used and results achieved.

2. Try to expand the literature review including some recent works (of last 3-years) in the similar field.

3. Figures are not shown in the paper. Only figure numbers are included. Why?

4. The validation of results is missing. Kindly validate the results either by hardware setup or at least by real-time HIL emulator.

5. Kindly analyze the performance under dynamic conditions.

6. Kindly rewrite the conclusion. Only novelty and originality of the work should be included in the conclusion. Kindly add 1-2 lines of future work in the last of conclusion section.

7. References are very old. Kindly cite at least 4 to 5 recent papers to validate that your work is novel and attempted by you first time in this domain.

8. Authors have reference [40] in the text but it has not been given in the reference list.

Reviewer #3: 1. Figures are not incorporated in the manuscript. Plz check.

2.Why author is considered sepic converter.

3.The abstract should be revised. Some parts of the abstract are unnecessary and should be removed. The main contributions should be highlighted in the abstract. Addressing some quantitative findings compared to available research works in the abstract is suggested.

4. How were the parameters of the proposed method selected?

5. Please validate the proposed method using load variation, and disturbances.

6. To show the efficacy of the proposed control method statistical analysis is needed. Please add it .

7.Please cite some paper https://doi.org/10.1002/oca.2798, 10.1109/JSYST.2020.3020275, https://doi.org/10.1049/iet-gtd.2018.5019,https://doi.org/10.3390/electronics11060927 etc

https://doi.org/10.1002/2050-7038.2824 etc.

8.Provide proof of the stability and convergence properties of the Proposed algorithm under various assumptions.

9.A major issue is the experimental analysis or real time work.

10.The contribution is in general not so highlighted area.

11.How the different PSC conditions are arrived at merits further explanation in the text,

12.If possible, Table should include the standard deviation (or variance for that matter) of power tracked after the MPP is settled (in the steady-state) to quantify the amount of oscillations

13.Also, the authors applied shading patterns for GP at the middle and at the end of P-V curve , it is better to add a shading pattern case for GP located at the beginning.

14.Conclusions section must be improved, underlining the relevant contributions and results and also future scope.

Reviewer #4: There is no figures in the paper. The authors are very careless while uploading the paper. However, the novelty of the paper is alos poor. It needs complete update. The results are not propely presented. The analysis is not done propely.

6. PLOS authors have the option to publish the peer review history of their article (what does this mean?). If published, this will include your full peer review and any attached files.

Reviewer #1: No

Reviewer #2: No

Reviewer #3: No

Reviewer #4: No

---

## [Author Response · Author response to Decision Letter 0]

1 Aug 2024

Response to Reviewer(s)' Comments

We would like to sincerely thank all the reviewers for their comments and suggestions, which help us to enhance our paper greatly. All the comments and suggestions from the reviewers have been considered and responded to carefully. The main changes carried out in the original submission according to the suggestions and comments of the reviewers are shown in blue in the revised version. The detailed responses to the reviewers are given as follows.

Reviewer #1: - For some reasons, Figures are not included in the final PDF file. So, I am unable to comment on results.

Comment 1: The work is limited to simulation. Mention, to which extent the designed optimization scheme is deployable on a real PV system? What are the challenges involved when the control law is to be implemented on a real system?

Response 1: The maximum power point is not oscillated around by metaheuristic-based MPPT tactics, in contrast to traditional MPPT techniques like P&O. The computational behaviour of metaheuristic optimisation algorithms is greatly influenced by their control settings. Depending on the nature of the situation, different control parameter values are appropriate. It is necessary to determine the appropriate control parameter values for every optimisation task. It has been discovered by experience that a lengthy process of trial and error is necessary to identify the ideal combination for maximum performance. However, just two parameters need to be adjusted for AOA. Programming requires less work overall. It is anticipated that the conventional microcontroller will provide easy implementation of the algorithm. 

Comment 2: Update the literature review by including notable works on MPPT control such as 10.3390/en16135039, 10.1371/journal.pone.0260480 and 10.3390/app12062773.

Response 2: Thank you for pointing this out. Changes have been made in the Introduction section. The details of the related reference papers are included in the reference section of the revised manuscript.

Comment 3: Please write quantitatively especially in the Abstract: How much 'quick' is quick response? How much 'excellent' is excellent?

Response 3: Thank you for pointing this out. Authors agree with this comment and the abstract has been modified in the revised paper. 

Comment 4: Include more rigorous analysis based on ITAE, IAE, ISE etc.

Response 4: The comments of the Reviewer are taken into account with gratitude to improve the quality of the manuscript with more sufficient result analysis. 

Condition ITAE IAE ISE

NUC1 1.6006e+06 1.9773e+07 5.0989e+09

NUC2 4.2266e+06 2.8296e+07 5.6235e+09

NUC3 2.5880e+06 2.4868e+07 6.5006e+09

Comment 5: Explicitly mention novelty of the proposed optimization technique.

Response 5: Thank you for your comment. It is a major problem to correctly track the GMPP in extreme weather circumstances, and traditional MPPT approaches are likely to be caught in between LMPPs and ineffectual. Using optimisation algorithms is a useful strategy for tracking GMPP in non-uniform solar irradiation conditions. Therefore, AOA-MPPT method under uniform and non-uniform conditions are investigated in this paper. In this method, the oscillations at steady state are reduced, which reduces the power loss. It requires less time to converge.

Comment 6: Avoid referring to waveforms/curves in a graph from their color. e.g. The 'green dotted line outlines the ideal ..."

Response 6: Thank you for your valuable comment. Correction have been carried out in the respective section of the revised manuscript.

Reviewer #2: 

Comment 1. Abstract is ambiguous, kindly arrange the abstract which can contain background of the work, objective, method used and results achieved.

Response 1: Thank you for pointing this out. Authors agree with this comment and the abstract has been modified in the revised paper.

Comment 2: Try to expand the literature review including some recent works (of last 3-years) in the similar field.

Response 2: As suggested by the reviewer, the latest literature related to MPPT is included in the Introduction section of the revised manuscript. The details of the related reference papers are included in the reference section of the revised manuscript.

Comment 3. Figures are not shown in the paper. Only figure numbers are included. Why?

Response 3: Thank you for your valuable comment. Correction have been carried out in the all sections of the revised manuscript.

Comment 4. The validation of results is missing. Kindly validate the results either by hardware setup or at least by real-time HIL emulator.

Response 4: Thank you for your valuable comment. Experimental setup and results included in the revised manuscript.

Comment 5. Kindly analyze the performance under dynamic conditions.

Response 5: Thank you for your comment. An effective AOA-MPPT algorithm may quickly and accurately converge to the necessary power, regardless of a gradual or abrupt change in sun irradiation. The maximum power obtained, settling time, and Tracking efficiency of three MPPT approaches are compared and depicted in Figure 15-19. According to the simulation results, it is concluded that the AOA-MPPT outperforms in terms of faster convergence to GP, higher efficiency, and less oscillations. The efficacy of the proposed MPPT method is confirmed for both partial shaded conditions and rapid changing irradiance conditions. Comparative results show that the proposed method exhibits greater performance than other methods under steady state and dynamic conditions.

Comment 6. Kindly rewrite the conclusion. Only novelty and originality of the work should be included in the conclusion. Kindly add 1-2 lines of future work in the last of conclusion section.

Response 6: Thank you for your comment. As suggested, conclusion section has been modified in the revised manuscript.

Comment 7. References are very old. Kindly cite at least 4 to 5 recent papers to validate that your work is novel and attempted by you first time in this domain.

Response 7: As suggested by the reviewer, the latest literature related to MPPT is included in the Introduction section of the revised manuscript. The details of the related reference papers are included in the reference section of the revised manuscript.

Comment 8. Authors have reference [40] in the text but it has not been given in the reference list.

Response 8: Thank you for pointing this out. Changes have been made in the revised manuscript.

Reviewer #3: 

Comment 1. Figures are not incorporated in the manuscript. Plz check.

Response 1: Thank you for pointing this out. Changes have been made in the revised manuscript.

Comment 2. Why author is considered sepic converter.

Response 2: Thank you for your comment. In this paper, SEPIC which has a high efficiency coefficient, has been selected. Even though it's a fourth-order electronic circuit, it has a lot of benefits that make it ideal for photovoltaic applications. These benefits include DC output current, series capacitor isolation between the input and output sides, flexible output gain, and non-inverting DC output voltage. The SEPIC may step up or step down the input voltage and operate in both buck and boost modes. It can thus track the maximum power point of the PV system at both the PV power level and input voltage. Despite having two inductors, the SEPIC converter's input ripple current is decreased, which lowers the peak inductor current and lowers the inductors' losses. These properties make SEPIC converter a suitable candidate for PV applications. 

Comment 3. The abstract should be revised. Some parts of the abstract are unnecessary and should be removed. The main contributions should be highlighted in the abstract. Addressing some quantitative findings compared to available research works in the abstract is suggested.

Response 3: Thank you for your comment. The findings demonstrate that AOA can follow MPP in 200–300ms under a variety of environmental changes. Moreover, the suggested AOA-MPPT has a tracking efficiency of almost 99% and settling time of 200-300ms. Besides, it has the ability to effectively manage the situation of partial shading. When it comes to tracking capabilities, transient behaviour, and convergence, AOA performs better than both P&O and GWO. 

Comment 4. How were the parameters of the proposed method selected?

Response 4: Thank you for pointing this out. The two parameters are μ and α. The results of this research indicate that whereas α is a sensitive parameter that specifies the exploitation accuracy across the iterations and is fixed at 5, μ is a control parameter that adjusts the search process and is fixed at 0.5.

Comment 5. Please validate the proposed method using load variation, and disturbances.

Response 5: Thank you for your comment. As suggested, a new figure is included in the results and discussion section of the revised manuscript to describe the effect of load variation on proposed method. When load has been reduced, the maximum power across the load also reduced. The proposed MPPT algorithm effectively handle the load variations.

Figure 20. Output power due to load variations

Comment 6. To show the efficacy of the proposed control method statistical analysis is needed. Please add it.

Response 6: The comments of the Reviewer are taken into account with gratitude to improve the quality of the manuscript with more sufficient result analysis.

Condition ITAE IAE ISE

NUC1 1.6006e+06 1.9773e+07 5.0989e+09

NUC2 4.2266e+06 2.8296e+07 5.6235e+09

NUC3 2.5880e+06 2.4868e+07 6.5006e+09

Comment 7. Please cite some paper https://doi.org/10.1002/oca.2798, 10.1109/JSYST.2020.3020275, https://doi.org/10.1049/iet-gtd.2018.5019, https://doi.org/10.3390/electronics11060927 etc https://doi.org/10.1002/2050-7038.2824 etc.

Response 7: Thank you for pointing this out. Changes have been made in the Introduction section. The details of the related reference papers are included in the reference section of the revised manuscript.

Comment 8. Provide proof of the stability and convergence properties of the Proposed algorithm under various assumptions.

Response 8: Thank you for your comment. An effective AOA-MPPT algorithm may quickly and accurately converge to the necessary power, regardless of a gradual or abrupt change in sun irradiation. The maximum power obtained, settling time, and Tracking efficiency of three MPPT approaches are compared and depicted in Figure 15-17. According to the simulation results, it is concluded that the AOA-MPPT outperforms in terms of faster convergence to GP, higher efficiency, and less oscillations.

Comment 9. A major issue is the experimental analysis or real time work.

Response 9: Thank you for your valuable comment. Experimental setup and results included in the revised manuscript.

Comment 10. The contribution is in general not so highlighted area.

Response 10: Thank you for your comment. In this paper, AOA-MPPT is proposed to track maximum power from PV under uniform and nonuniform conditions. Because AOA algorithms explore a large search space, AOA lower the likelihood of adhering to local maxima. Furthermore, this is less expensive computationally than AI methods. With minimal steady-state oscillations, the suggested AOA-MPPT technique can track GMPP with extremely high tracking speed and efficiency. 

Comment 11. How the different PSC conditions are arrived at merits further explanation in the text.

Response 11: Thank you for your comment. Three Non-Uniform Conditions have been chosen in which GMPP is located in middle, end and start positions respectively.

Comment 12. If possible, Table should include the standard deviation (or variance for that matter) of power tracked after the MPP is settled (in the steady-state) to quantify the amount of oscillations.

Response 12: Thank you for your comment. Once the steady state is reached, the AOA-MPPT algorithm continues to maintain the MPP with almost zero fluctuation, as shown in Figure 10.

 Figure 10 Maximum power delivered to the load for NUC-1

Comment 13. Also, the authors applied shading patterns for GP at the middle and at the end of P-V curve, it is better to add a shading pattern case for GP located at the beginning.

Response 13: Thank you for pointing out. As suggested, shading pattern case for GP located at the beginning has been included in the revised manuscript. When it comes to tracking capabilities, transient behaviour, and convergence, AOA performs better than both P&O and GWO. These instances lead to the conclusion that the samples' MPP convergence is mostly independent of their starting positions.

(a) 

(b) 

(c)

Figure 8. PV characteristics under NUCs with different GP positions

Comment 14. Conclusions section must be improved, underlining the relevant contributions and results and also future scope.

Response 14: Thank you for your comment. As suggested, conclusion section has been modified in the revised manuscript.

Reviewer #4: There is no figures in the paper. The authors are very careless while uploading the paper. However, the novelty of the paper is also poor. It needs complete update. The results are not properly presented. The analysis is not done properly.

Response: Thank you for your comment. Corrections have been made in the revised manuscript. In this paper, AOA-MPPT is proposed to track maximum power from PV under uniform and nonuniform conditions. Because AOA algorithms explore a large search space, AOA lower the likelihood of adhering to local maxima. Furthermore, this is less expensive computationally than AI methods. With minimal steady-state oscillations, the suggested AOA-MPPT technique can track GMPP with extremely high tracking speed and efficiency. The proposed AOA-MPPT's efficacy under different insolation patterns has been validated using three nonuniform conditions in terms of convergence, tracking speed, steady state oscillations, and tracking efficiency. The tracking efficiency of the AOA-MPPT is above 99% and settling time is 200 to 300ms for all three non-uniform conditions.

The authors are very grateful to the editor and reviewers for all their careful assessments and constructive suggestions, which have helped to improve the presentation and enhance the quality of the research paper. The above Response sheet is also attached with the revised manuscript.

---

## [Decision Letter · Decision Letter 1]

12 Aug 2024

PONE-D-24-06244R1Arithmetic Optimization based MPPT for photovoltaic systems operating under non-uniform situationsPLOS ONE

Dear Dr.Omar,

Thank you for submitting your manuscript to PLOS ONE. After careful consideration, we feel that it has merit but does not fully meet PLOS ONE’s publication criteria as it currently stands. Therefore, we invite you to submit a revised version of the manuscript that addresses the points raised during the review process.

**ACADEMIC EDITOR:** The reviewers recommend reconsideration the manuscript with revision and modification. I invite the authors to resubmit the manuscript after addressing the comments raised by the reviewers.

Please submit your revised manuscript by Sep 26 2024 11:59PM.  If you will need more time than this to complete your revisions, please reply to this message or contact the journal office at plosone@plos.org. Please include the following items when submitting your revised manuscript:A rebuttal letter that responds to each point raised by the academic editor and reviewer(s). You should upload this letter as a separate file labeled 'Response to Reviewers'.A marked-up copy of your manuscript that highlights changes made to the original version. You should upload this as a separate file labeled 'Revised Manuscript with Track Changes'.An unmarked version of your revised paper without tracked changes. You should upload this as a separate file labeled 'Manuscript'.If applicable, we recommend that you deposit your laboratory protocols in protocols.io to enhance the reproducibility of your results. Protocols.io assigns your protocol its own identifier (DOI) so that it can be cited independently in the future. For instructions see: https://journals.plos.org/plosone/s/submission-guidelines#loc-laboratory-protocols. Additionally, PLOS ONE offers an option for publishing peer-reviewed Lab Protocol articles, which describe protocols hosted on protocols.io. Read more information on sharing protocols at https://plos.org/protocols?utm_medium=editorial-email&utm_source=authorletters&utm_campaign=protocols.

We look forward to receiving your revised manuscript.

Kind regards,

Dhanamjayulu C, Ph.D & Post.Doc

Academic Editor

PLOS ONE

Journal Requirements:

Additional Editor Comments:

The reviewers recommend reconsideration the manuscript with revision and modification. I invite the authors to resubmit the manuscript after addressing the comments raised by the reviewers.

Reviewers' comments:

Reviewer's Responses to Questions

**Comments to the Author**

1. If the authors have adequately addressed your comments raised in a previous round of review and you feel that this manuscript is now acceptable for publication, you may indicate that here to bypass the “Comments to the Author” section, enter your conflict of interest statement in the “Confidential to Editor” section, and submit your "Accept" recommendation.

Reviewer #1: All comments have been addressed

Reviewer #3: All comments have been addressed

2. Is the manuscript technically sound, and do the data support the conclusions?

Reviewer #1: Yes

Reviewer #3: Partly

3. Has the statistical analysis been performed appropriately and rigorously? 

Reviewer #1: I Don't Know

Reviewer #3: No

4. Have the authors made all data underlying the findings in their manuscript fully available?

Reviewer #1: Yes

Reviewer #3: Yes

5. Is the manuscript presented in an intelligible fashion and written in standard English?

Reviewer #1: Yes

Reviewer #3: Yes

6. Review Comments to the Author

Reviewer #1: Authors have addressed all the comments suggested. The revised version of the paper has been significantly improved. The paper can be accepted in its present form.

Reviewer #3: 1. So many comments are not answer properly. like Q5, 6 and 8, 12.

2. Please read some paper for answer these questions. 10.1109/TIA.2024.3413052, and 10.1109/TIA.2023.3321031 .

3. Follow these papers how the stability analysis and convergence characteristics are analysis and also cite these.

4. please take more case studies to analysis and verify the proposed algorithm .

5. The future scope should be board. not like a comment.

6. The statical analysis is not correct.

7. Experimental results are not correct. Please take proper scale and give in paper.

7. PLOS authors have the option to publish the peer review history of their article (what does this mean?). If published, this will include your full peer review and any attached files.

Reviewer #1: No

Reviewer #3: No

---

## [Author Response · Author response to Decision Letter 1]

4 Sep 2024

Response to Reviewer(s)' Comments

We would like to sincerely thank all the reviewers for their comments and suggestions, which help us to enhance our paper greatly. All the comments and suggestions from the reviewers have been considered and responded to carefully. The main changes carried out in the original submission according to the suggestions and comments of the reviewers are shown in blue in the revised version. The detailed responses to the reviewers are given as follows.

Reviewer #1: Authors have addressed all the comments suggested. The revised version of the paper has been significantly improved. The paper can be accepted in its present form.

Response: Thank you for your valuable comment.

Reviewer #3: 

Comment 1. So many comments are not answer properly. like Q5, 6 and 8, 12.

Response 1: The comments of the Reviewer are taken into account with gratitude to improve the quality of the manuscript with more sufficient result analysis. Statistical analysis and experimental results are included in the revised manuscript.

Table 3 Numerical Results obtained from AOA and GWO method

Algorithm Condition MAE MSE RMSE Mean SD SD in steady state

AOA NUC1 34.5 8.9031e+03 94.35 603.2080 0.0459 0.0029

 NUC2 108.0709 2.1478e+04 146.55 441.9291 0.1368 0.0316

 NUC3 28.7572 7.5173e+03 86.7026 711.6504 0.0310 0.0019

GWO NUC1 45.6 9.686e+03 98.42 599.3480 0.052 0.0042

 NUC2 120.34 2.3311e+04 152.68 420.5824 0.142 0.0426

 NUC3 33.26 8.619e+03 92.84 700.7824 0.036 0.0026

The statistical terms such as, mean, standard deviation (SD), mean absolute error (MAE), mean square error (MSE), and root mean square error (RMSE) are used to analyse the results under non uniform conditions, as illustrated in Table 3. In addition, standard deviation of power tracked after the MPP is settled (in the steady-state) is tabulated to quantify the amount of oscillations.

According to Figure 18a, the convergence time with GWO-MPPT for UC is 280ms and it can be reduced to 160ms with the aid of AOA-MPPT. Also, the convergence time is reduced to 120ms with AOA and 150ms with GWO for change in irradiance condition from UC to NUC1.

Figure 20 is included to describe the effect of load variation on proposed method. When resistive load has been reduced from 40Ω to 20Ω, the disturbance occurs in output voltage of converter but the AOA tracks the actual power within 10ms.

Comment 2. Please read some paper for answer these questions. 10.1109/TIA.2024.3413052, and 10.1109/TIA.2023.3321031.

Response 2: Thank you for pointing this out. Changes have been made in the respective section. The details of the related reference papers are included in the reference section of the revised manuscript.

Comment 3. Follow these papers how the stability analysis and convergence characteristics are analysis and also cite these.

Response: Thank you for pointing this out. Changes have been made in the results section. The details of the related reference papers are included in the reference section of the revised manuscript. 

According to Figure 18a, the convergence time with GWO-MPPT for UC is 280ms and it can be reduced to 160ms with the aid of AOA-MPPT. Also, the convergence time is reduced to 120ms with AOA and 150ms with GWO for change in irradiance condition from UC to NUC1.

Comment 4. please take more case studies to analysis and verify the proposed algorithm .

Response: Thank you for your comment. Three non-uniform insolation conditions with different GP positions and two different fast irradiance varying conditions have been analysed. An effective AOA-MPPT algorithm may quickly and accurately converge to the necessary power, regardless of a gradual or abrupt change in sun irradiation. The maximum power obtained, settling time, and Tracking efficiency of three MPPT approaches are compared and depicted in Figure 15-19.

Comment 5. The future scope should be broad. not like a comment.

Response: Thank you for your comment. As suggested, conclusion section has been modified with future scope in the revised manuscript.

In light of these findings, the work's next stage will involve hardware validation of the suggested AOA-MPPT algorithm for PV systems operating in uneven and quickly changing settings. It is anticipated that the PV community, comprising researchers and practitioners alike, will be highly interested in our work. In order to build a solar photovoltaic grid-connected power generating system in the future, authors will incorporate the suggested AOA-MPPT technique with PV inverters, with the goal of enhancing the overall energy harvesting efficiency. Analysis can be done on any single switch DC-DC boost converter with numerous peaks in the P-V curve can use the AOA-MPPT. In addition, the authors will take into account a practical solution to address the issue of partially shadowing the PV array surface.

Comment 6. The statical analysis is not correct.

Response 6: The comments of the Reviewer are taken into account with gratitude to improve the quality of the manuscript with more sufficient result analysis.

Table 3 Numerical Results obtained from AOA and GWO method

Algorithm Condition MAE MSE RMSE Mean SD SD in steady state

AOA NUC1 34.5 8.9031e+03 94.35 603.2080 0.0459 0.0029

 NUC2 108.0709 2.1478e+04 146.55 441.9291 0.1368 0.0316

 NUC3 28.7572 7.5173e+03 86.7026 711.6504 0.0310 0.0019

GWO NUC1 45.6 9.686e+03 98.42 599.3480 0.052 0.0042

 NUC2 120.34 2.3311e+04 152.68 420.5824 0.142 0.0426

 NUC3 33.26 8.619e+03 92.84 700.7824 0.036 0.0026

The statistical terms such as, mean, standard deviation (SD), mean absolute error (MAE), mean square error (MSE), and root mean square error (RMSE) are used to analyse the results under non uniform conditions, as illustrated in Table 3. In addition, standard deviation of power tracked after the MPP is settled (in the steady-state) is tabulated to quantify the amount of oscillations.

Comment 7. Experimental results are not correct. Please take proper scale and give in paper.

Response: Thank you for your valuable comment. Experimental setup and results included in the revised manuscript.

The authors are very grateful to the editor and reviewers for all their careful assessments and constructive suggestions, which have helped to improve the presentation and enhance the quality of the research paper. The above Response sheet is also attached with the revised manuscript.

 EXPERIMENTAL RESULTS

The experimental setup for the proposed system is developed according to the simulation specification as shown in Figure 21. The proposed control algorithm for the DC-DC converter is implemented using PIC 16F877A from Microchip. 

Figure 21. Experimental Setup for the Proposed PV system

(a)

(b)

(c)

(d)

Figure 22. Experimental Results (a) Control signal from MPPT controller (b) Converter output under UC

Figure 22 shows the experimental results for proposed solar PV system under UC where the results obtained are similar to the results obtained from simulation. Figure 22a shows the control signal obtained from the proposed AOA-MPPT algorithm to the switches which also shows the voltage across the diode. Figure 22b shows the converter’s steady state output voltage of 198 V and current of 3.74 A when NUC-3: Insolation = [1000 600 1000 1000] W\\/m^2 and Temperature = 25°C. Figure 22c depicts the variation of voltage and current during non-uniform environmental conditions described in NUC-3. Figure 22.d. depicts the variation of voltage and current during fast changing solar irradiance condition. The experimental results evidences the efficacy of proposed AOA-algorithm under different circumstances and closely matches with simulation results.

---

## [Decision Letter · Decision Letter 2]

16 Sep 2024

Arithmetic Optimization based MPPT for photovoltaic systems operating under nonuniform situations

PONE-D-24-06244R2

Dear Dr. Ahmed I. Omar,

We’re pleased to inform you that your manuscript has been judged scientifically suitable for publication and will be formally accepted for publication once it meets all outstanding technical requirements.

Kind regards,

Dhanamjayulu C, Ph.D & Post.Doc

Academic Editor

PLOS ONE

Additional Editor Comments (optional):

The authors have addressed the reviewers’ comments properly

The article can be accepted for the publication in present form

Reviewers' comments:

Reviewer's Responses to Questions

**Comments to the Author**

1. If the authors have adequately addressed your comments raised in a previous round of review and you feel that this manuscript is now acceptable for publication, you may indicate that here to bypass the “Comments to the Author” section, enter your conflict of interest statement in the “Confidential to Editor” section, and submit your "Accept" recommendation.

Reviewer #1: All comments have been addressed

Reviewer #3: All comments have been addressed

2. Is the manuscript technically sound, and do the data support the conclusions?

Reviewer #1: Yes

Reviewer #3: Partly

3. Has the statistical analysis been performed appropriately and rigorously? 

Reviewer #1: Yes

Reviewer #3: Yes

4. Have the authors made all data underlying the findings in their manuscript fully available?

Reviewer #1: Yes

Reviewer #3: Yes

5. Is the manuscript presented in an intelligible fashion and written in standard English?

Reviewer #1: Yes

Reviewer #3: Yes

6. Review Comments to the Author

Reviewer #1: Authors have addressed all the suggested changes. The revised paper has been significantly improved and is recommended for acceptance.

Reviewer #3: The author try to improved the manuscript but still main points are lagging which can increase the value of the paper like stability analysis, PSCs . Also Fig 14 and 20.

7. PLOS authors have the option to publish the peer review history of their article (what does this mean?). If published, this will include your full peer review and any attached files.

Reviewer #1: No

Reviewer #3: No

---

## [Editor Report · Acceptance letter]

17 Oct 2024

PONE-D-24-06244R2 

PLOS ONE

Dear Dr. Omar, 

I'm pleased to inform you that your manuscript has been deemed suitable for publication in PLOS ONE. Congratulations! Your manuscript is now being handed over to our production team.

Kind regards, 

on behalf of

Dr. Dhanamjayulu C 

Academic Editor

PLOS ONE